# Extrapolating Continuous Vegetation Water Content To Understand Sub-daily Backscatter Variations

Paul C. Vermunt[1], Susan C. Steele-Dunne[1], Saeed Khabbazan[1], Jasmeet Judge[2], and Nick C. van de Giesen[1]

[1]Faculty of Civil Engineering and Geosciences, Delft University of Technology, 2628 CN Delft, The Netherlands
[2]Center for Remote Sensing, Agricultural and Biological Engineering Department, University of Florida, Gainesville, FL 32611 USA.

**Correspondence:** Paul C. Vermunt (p.c.vermunt@tudelft.nl)

**Abstract.** Microwave observations are sensitive to vegetation water content (VWC). Consequently, the increasing temporal and spatial resolution of spaceborne microwave observations creates a unique opportunity to study vegetation water dynamics and its role in the diurnal water cycle. However, we currently have a limited understanding of sub-daily variations in VWC and how they affect microwave observations. This is partly due to the challenges associated with measuring internal VWC for validation, particularly non-destructively and at timescales of less than a day. In this study, we aimed to (1) use field sensors to reconstruct diurnal and continuous records of internal VWC of corn, and (2) use these records to interpret the sub-daily behaviour of a 10-day time series of polarimetric L-band backscatter with high temporal resolution. Sub-daily variations of internal VWC were calculated based on the cumulative difference between estimated transpiration and sap flow rates at the base of the stems. Destructive samples were used to constrain the estimates and for validation. The inclusion of continuous surface canopy water estimates (dew or interception) and surface soil moisture allowed us to attribute hour-to-hour backscatter dynamics to either internal VWC, surface canopy water or soil moisture variations. Our results showed that internal VWC varied with 10-20% during the day in non-stressed conditions, and the effect on backscatter was significant. Diurnal variations of internal VWC and nocturnal dew formation affected vertically polarized backscatter most. Moreover, multiple linear regression suggested that the diurnal cycle of VWC on a typical dry day leads to a two (HH- and cross-pol) to almost four (VV-pol) times higher diurnal backscatter variation than the soil moisture drydown does. These results demonstrate that radar observations have the potential to provide unprecedented insight into the role of vegetation water dynamics in land-atmosphere interactions at sub-daily timescales.

## 1 Introduction

The long heritage of research on remote soil moisture and biophysical parameter retrieval has shown that backscatter is sensitive to dielectric properties of vegetation, which is strongly related to its water content (Konings et al. (2019); Steele-Dunne et al. (2017)). For a long time, this sensitivity to vegetation water content (VWC), here defined as the weight of water captured inside the plant material above a square meter of ground [kg m$^{-2}$], was considered a barrier to soil moisture retrieval. In the last decade however, backscatter sensitivity to VWC has been used for studies on plant hydraulics and water stress in agriculture

and ecosystems (e.g. Frolking et al. (2011); Steele-Dunne et al. (2012); Schroeder et al. (2016); Emmerik et al. (2017); Konings et al. (2017); Steele-Dunne et al. (2019); El Hajj et al. (2019)).

The increasing temporal and spatial resolution of spaceborne radar observations creates opportunities for more detailed and extensive (eco)hydrological studies. In addition to the frequent C-band Synthetic Aperture Radar (SAR) observations from Sentinel-1 (Torres et al. (2012)) and the Radar Constellation Mission (Thompson (2015)), other frequencies such as the L- and S-band mission NISAR (launch planned in 2023), the L-band mission ROSE-L (2028), and the P-band mission BIOMASS (2023) will be available within the next decade (Rosen et al. (2017); Pierdicca et al. (2019); Quegan et al. (2019)). Moreover, commercial providers such as CapellaSpace and Iceye are building satellite constellations with X-band instruments (Farquharson et al. (2021); Ignatenko et al. (2020)). These constellations will ensure multiple observations per day. As a result, the availability of spaceborne backscatter observations in the near future will offer a unique possibility to study vegetation water dynamics on different spatio-temporal scales.

However, we currently lack crucial knowledge on backscatter sensitivity to vegetation water dynamics. Soil moisture retrieval algorithms, for example, generally consider the confounding effects of vegetation water as time invariant or seasonally variant only (Kim et al. (2017)). Well-established electromagnetic models have been developed and calibrated based on seasonally variant VWC only (e.g. Bracaglia et al. (1995)). Moreover, the effect of surface canopy water (SCW), i.e. dew or rainfall interception, is also usually ignored (Vermunt et al. (2020); Xu et al. (2021)). The omission of sub-daily vegetation water dynamics causes potential retrieval errors (Brancato et al. (2017)), and more importantly, hinders our understanding of the extent to which radar backscatter could be used to monitor vegetation water dynamics. Without this knowledge, the upcoming spaceborne observations cannot be used to their full potential.

Several studies have related observed diurnal backscatter cycles to vegetation water dynamics. Clear diurnal cycles were found in tower-based observations from forest stands (e.g. Hamadi et al. (2014); Monteith and Ulander (2021)) and agricultural cropland (e.g. Vermunt et al. (2020)), as well as in aggregated satellite observations from larger forested areas (Paget et al. (2016); Emmerik et al. (2017); Konings et al. (2017)). These studies have made important contributions to the understanding of sub-daily backscatter behaviour. However, a persistent challenge is the lack of in-situ data for ground-truth validation. In-situ soil moisture can be routinely measured using a variety of sensors (Dobriyal et al. (2012); Cosh et al. (2016)). Surface canopy water can be measured continuously using leaf wetness sensors (Cosh et al. (2009); Vermunt et al. (2020)). However, internal VWC is still generally measured using laborious destructive sampling, particularly in agricultural fields (e.g. Vreugdenhil et al. (2018); Emmerik et al. (2015); Ye et al. (2021)). This is acceptable for monitoring seasonal changes, but is prohibitively time-consuming and labor-intensive for sub-daily variations. Hence, it is crucial to find a more efficient way to obtain continuous, quantitative estimates of sub-daily VWC variations.

For woody constituents in trees, dendrometers have been used to infer water content non-destructively after detrending, and similarly, reflectometry (TDR and FDR) and capacitance-style sensors have been used to derive water content indirectly by measuring dielectric permittivity (Konings et al. (2021)). Moreover, a water balance-style approach using sap flow sensors have been used by the tree physiology community to estimate diurnal changes in tree stem water storage (Goldstein et al. (1998); Meinzer et al. (2004); Čermák et al. (2007); Phillips et al. (2008); Köcher et al. (2013)).

The objectives of this study were to test the potential of this non-destructive sap flow approach for estimating sub-daily VWC variations in herbaceous plants, and to use these estimates to better understand what controls sub-daily variations of L-band backscatter. Specifically, we adapted this sap flow methodology, described in section 2, to estimate 15-minute changes in corn VWC using sap flow sensors and a weather station. An extensive data set from a field campaign in the Netherlands in 2019 was used to evaluate the adapted method against diurnal cycles of VWC obtained by destructive sampling. Finally, the technique was applied to reconstruct sub-daily VWC variability of multiple consecutive days from another field campaign in Florida in 2018. In this campaign, high temporal resolution tower-based polarimetric L-band backscatter was collected. The reconstructed VWC was used, together with simultaneously collected soil moisture and surface canopy water (SCW), to gain better understanding of what controls sub-daily backscatter behaviour.

## 2 Estimating diurnal variations in tree water content using sap flow probes

Diurnal variations of internal VWC have been estimated in trees before, mainly in studies focused on understanding the functional role of stem water reserves on daily tree water use. A well-established *in situ* method uses sap flow probes at the base of the stem and in the crown (e.g. Goldstein et al. (1998); Meinzer et al. (2004); Čermák et al. (2007); Phillips et al. (2008); Köcher et al. (2013)). This method is based on the time lag between transpiration and basal sap flow, as a result of a tree's hydraulic capacitance, which is the change in water content per unit change in water potential (e.g. kg MPa$^{-1}$) (Goldstein et al. (1998); Oguntunde et al. (2004)). Morning transpiration, driven by the atmospheric evaporative demand, causes depletion of internal VWC in the crown, and, depending on the hydraulic capacitance, a drop in water potential. In response to the resulting potential gradient, sap flow rates increase to replenish the depleted VWC. As long as transpiration rates exceed basal sap flow rates, water is withdrawn from internal VWC, and when basal sap flow exceeds transpiration, internal VWC is refilled. Consequently, the diurnal variation in tree VWC could be calculated from the cumulative differences between basal sap flow and whole-crown transpiration (see the second term of equation 1).

$$VWC(t) = VWC(t_0) + \sum_{i=t_0}^{t}(F_i - T_i)\Delta t \tag{1}$$

, where $VWC(t)$ is the estimated VWC at time t, $VWC(t_0)$ is a reference VWC at t=0, $F$ is basal sap flow, $T$ is whole-crown transpiration, both in mass per unit of time, and $\Delta$t is the duration of a time step.

In these studies on trees, whole-crown transpiration was estimated from branch and basal sap flow, based on two assumptions. First, time lags between branch sap flow in the crown and transpiration were assumed to be negligible compared to time lags between branch and basal sap flow. Hence, the averaged daily cycles of sap flow in the monitored branches were assumed to approximate the cycles of whole-crown transpiration. Second, most studies assumed that the 24-hour sums of whole-crown transpiration and basal sap flow were equal (Goldstein et al. (1998); Čermák et al. (2007); Phillips et al. (2008); Köcher et al. (2013)). This assumption made it possible to estimate whole-crown transpiration rates by first dividing averaged branch sap

flow by its daily sum, and then multiplying by the daily sum of basal sap flow. The corresponding assumption is that all water
that is withdrawn from internal VWC is replaced within 24 hours.

## 3 Data and Methods

Section 3.1 relates to the adjustments and data required to make the sap flow approach (section 2) applicable to corn. Data from
a field campaign in The Netherlands in 2019 were used to evaluate the adjusted method. Section 3.2 relates to the methodology
and data used from our field campaign in Florida in 2018 for interpreting sub-daily backscatter behaviour.

### 3.1 Applying the sap flow approach to estimate diurnal variations in corn VWC

#### 3.1.1 Adjustments and evaluation of the sap flow approach

We investigated the potential of the sap flow method (section 2) for estimating diurnal VWC variations in corn plants. The
largest differences between corn plants and trees are related to hydraulic capacitance and structure. Corn plants have much
lower hydraulic capacitance than most trees (Langensiepen et al. (2009)), and hence shorter time lags between transpiration
and basal sap flow. As a consequence, installing a sap flow sensor as a surrogate for transpiration would be problematic, since
the assumption of negligible time lag between transpiration and upper sap flow, compared to the lag with basal sap flow, is
invalid. Moreover, transpiring corn leaves are somehow evenly distributed across the stem, in contrast to trees with a crown,
which makes the placement of a second sensor to represent transpiration nearly impossible. For these reasons, we estimated
transpiration using indirect estimates of reference evapotranspiration (ETo) instead. Details from sap flow measurements and
ETo estimates are given in section 3.1.2.

A widely used approach to derive transpiration from ETo is a linear conversion using crop factors, e.g. the FAO-56 dual
crop coefficient model Allen et al. (1998). However, in many cases, these estimations systematically over- or underestimate
direct observations of transpiration (Ding et al. (2013); Rafi et al. (2019)) or sap flow (Langensiepen et al. (2009)), while basal
sap flow and transpiration at the leaves must equal over a sufficiently long time period (Swanson (1994)). For our data sets,
Penman-Monteith derived transpiration (Allen et al. (1998)) is systematically lower than measured sap flow. Because sap flow
is our most direct measurement, we chose to estimate transpiration by rescaling ETo estimates using sap flow measurements.
This means that information on the diurnal shape of ETo is derived from the Penman-Monteith equation, and that these ETo
estimates are then scaled so that the resulting transpiration estimates are consistent with sap flow over a given period of time.
We tested three different approaches to rescale ETo estimates using sap flow measurements. The first approach was similar
to the rescaling of branch sap flow to whole-crown transpiration in trees, described in section 2. Transpiration was assumed
to equal basal sap flow during a 24 hour period, and 15-minute ETo estimates were divided by their 24-hour sum and then
multiplied by the 24-hour sum of basal sap flow (see eq. 2 in Table 1).

However, the assumption of complete replacement of withdrawn water within 24 hours may not always hold. This is for example the case when water accumulates as a result of growth, or when a plant is unable to replace the transpired water within a day as a result of stress. Therefore, we also tested the effect of relaxing this assumption, and using multiple days instead: 3, 5 or 7 consecutive days surrounding the day of interest, or all measured days in the data set. Both approaches assume a simple, linear relation between ETo and transpiration. It will be shown that this assumption can cause an offset between the timing of the diurnal cycles of sampled and reconstructed VWC. This issue was addressed by adopting the cumulative distribution function (CDF) matching method, previously used to rescale satellite-derived surface soil moisture to observations (Reichle and Koster (2004); Drusch et al. (2005); Brocca et al. (2011)). This non-linear approach removes systematic differences between two data sets by matching the CDF's of both data sets (Brocca et al. (2011)). Here, we matched the CDF's of the ETo and sap flow data. This was achieved by first ranking all 15-minute data from both data sets from low to high values, and then fitting a second-order polynomial function through the scatter plot of both ranked data sets. Subsequently, this function was used to convert the 15-min ETo data to transpiration estimates. CDF-matching was also performed for 1, 3, 5, 7 consecutive, and all available days. Fig. 4 illustrates CDF-matching and its results for three days of our data.

**Table 1.** The three tested approaches to estimate transpiration (T) using Penman-Monteith derived ETo estimates and sap flow measurements.

| Approach | assumptions | equations | |
|---|---|---|---|
| Linear-24h | withdrawn water is replaced within 24-hours. | $T(t) = ETo(t)\, F_{24h}/ETo_{24h\star}$ | (2) |
| | T is linearly related to ETo | | |
| Linear- multiple days | withdrawn water is replaced within $n$ days. | $T(t) = ETo(t)\, F_{nd}/ETo_{nd\star}$ | (3) |
| | T is linearly related to ETo | | |
| Nonlinear - CDF-matching | withdrawn water is replaced within $n$ days. | $T(t) = a^{\dagger}\, ETo(t) + b^{\dagger}\, ETo(t)^2$ | (4) |
| | CDF of T equals CDF of F | | |

$\star$subscripts $24h$ and $nd$ relate to the 24-hour sum and $n$-days sum, respectively.

$^{\dagger}$ $a$ and $b$ are found by a 2$^{nd}$ order polynomial fit through ranked F and ETo data, illustrated in Fig. 4(c) .

VWC samples obtained by destructive sampling during the 2019-campaign (section 3.1.2) were used to validate the method. For the selected days (Fig. 1), we used one of the five sampling times to constrain the daily cycle ($VWC(t_0)$ in eq. 1). The other four independent samples were compared against the estimated diurnal cycle of VWC variations. For each day, we calculated the Root Mean Square Error (RMSE) between the four independent samples and reconstructed VWC on these four times. All five samples were used as ($VWC(t_0)$) once to determine the best time to constrain the reconstruction.

In summary, we adapted and evaluated the sap flow methodology to estimate diurnal cycles of corn VWC through the following three steps.

① The diurnal cycle of transpiration was estimated from ETo and sap flow data, using three different approaches (Table 1).

② Sub-daily variations in VWC were estimated by calculating the cumulative difference between 15-minute basal sap flow and transpiration estimates (eq. 1).

③ The resulting estimates of diurnal VWC variations were compared against destructive measurements of VWC.

### 3.1.2 Experimental site and data collection

*Experimental site 2019*

The field campaign in The Netherlands was conducted in Reusel (51.319N , 5.173E), at Van den Borne Aardappelen. There, field corn was planted on a sandy soil with a density of 8 plants m$^{-2}$, and harvested for silage after the required senescence, 148 days after planting. The Netherlands has a temperate maritime climate. However, maximum national temperature records were broken close to the field site during the growing season of 2019, and it was the second anomalously dry summer in a row (Bartholomeus et al. (2020)).

*Sap flow*

Sap flow was monitored near the base of the stem using stem-flow gauges produced by Dynamax Inc. (Houston, TX, USA). The measurement is based on the stem heat balance theory (Sakuratani (1981)). A flexible collar strap with built-in heater strip and thermocouples is wrapped around a corn stem, about 20 cm above the ground, and then isolated and protected from environmental conditions such as rain and radiation. The entire circumference of the stem receives a constant heat input from the heater strip. As sap movement carries heat, thermal dissipation corresponds to the sap flow rate. Therefore, the change in temperature is used as a tracer for sap flow [g hr$^{-1}$], thereby taking into account the heat transfer to the stem tissue and the ambient air. Conversion to [mm 15-min$^{-1}$] was performed using the density of liquid water and the planting density. Because the collar straps are designed to fit a certain range of stem diameters, we collected data in mid- and late season.

In 2019, a maximum of two sensors were installed due to power limitations. Because one sensor failed, the used data is from a single sensor. Gaps in the time series were caused by disturbances in the connection with the battery.

*Reference evapotranspiration*

A weather station was installed on the edge of the experimental site, with a ECH2O Rain Model ECRN-100 rain gauge, Apogee SP-212 pyranometer (solar radiation), a Davis Cup anemometer (wind and gust speed, and wind direction), and a HOBO Temperature/RH Smart Sensor Model S-THB-M008 (temperature and relative humidity). Reference evapotranspiration (ETo) was estimated using the Penman-Monteith approach described by Zotarelli et al. (2010).

*Sampling*

Vegetation water content (VWC) was measured by destructive sampling. Six field-representative samples were taken from designated sampling areas. Any present dew or interception was removed with paper towels before the samples were weighted

to determine average fresh biomass per plant in kg ($m_f$). Samples were oven-dried at 60 °C for 4-8 days, depending on growth stage. These dried samples were weighed again to determine average dry biomass per plant in kg ($m_d$). Field-representative VWC [kg m$^{-2}$] was estimated by multiplying the evaporated water per plant [kg] with the number of plants per m$^2$ ($\rho_{plant}$), see equation 5.

$$VWC = (m_f - m_d)\rho_{plant} \qquad (5)$$

In 2019, we aimed to capture full diurnal cycles of VWC. Hence, we sampled on five equally distributed times between sunrise and sunset, on 12 days spread throughout the season. Seasonal VWC variations were monitored by predawn sampling only.

Figure 1 shows the availability of the data required to evaluate the adjusted methodology for estimating 15-minute VWC variations. The availability of sap flow, ETo, and VWC sampling data matched on July 25, August 23, and August 28.

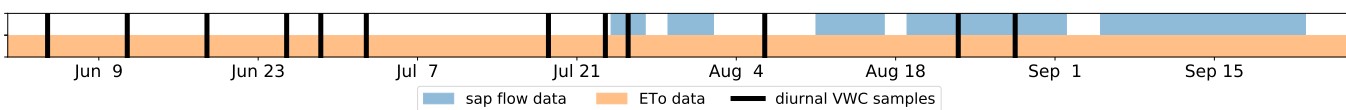

**Figure 1.** Availability of the data required to evaluate the adjusted methodology for estimating 15-minute VWC variations. The availability of sap flow, ETo, and sampling data matched on July 25, August 23, and August 28.

*Surface canopy water and soil moisture*

Measurements of surface canopy water (dew, interception) and root zone soil moisture were used as ancillary data sets, to support the evaluation of the reconstructed VWC estimations. Surface canopy water (SCW) was monitored using PHYTOS-31 Leaf Dielectric Wetness sensors. Three sensors were installed on different heights in the vegetation layer, and one sensor failed during the season. Measured leaf areas were used to convert sensor output to full-canopy SCW [kg m$^{-2}$]. Details of this conversion and sensor properties are described in Vermunt et al. (2020).

Soil moisture ($\theta$) was observed in two pits with 15-minute resolution, using EC-5 sensors on 5, 10, 20, 40 and 80 cm depth. These measurements were averaged based on depth. Root zone soil moisture was estimated by integrating the measurements from all depth over a soil column of 100 cm, based on the thickness of the soil layer associated with the depth of the sensor.

## 3.2   Interpreting the behaviour of sub-daily L-band backscatter

### 3.2.1   Approach and data requirement

To gain better understanding of what controls sub-daily L-band backscatter behaviour, we analyzed tower-based observations using continuous time series of the three moisture stores in the corn field: (1) VWC, (2) SCW, and (3) surface soil moisture ($\theta$).

Details of the collection of these time series are given in section 3.2.2. The longest period for which we had all data available
was from June 4 00:00 to June 13 10:15. During this period, the corn is at maximum height and LAI, and 1-2 weeks before
harvest on June 18. All analyses were conducted for this period.

Insights in the separate effects of the three different moisture stores on sub-daily backscatter ($\sigma^0$) variations were gained
by quantifying their relations through multiple linear regression. The relation between sub-daily backscatter variations and
changes in these dynamic moisture stores was described by:

$$\sigma^0(t) = \sigma_{t0}^0 + a(\theta_t - \theta_{t0}) + b(VWC_t - VWC_{t0}) + c(SCW_t - SCW_{t0}) \tag{6}$$

,where $t_0$ is the first radar acquisition time of the day (01:00), and assuming linear relations between $\sigma^0$ and the individual
moisture stores. The regression coefficients $a$ $[dB/m^3m^{-3}]$, $b$ $[dB/kgm^{-2}]$, and $c$ $[dB/kgm^{-2}]$ were used to quantify the
change in backscatter within a day as a result of change in moisture, and were derived for each polarization separately.

### 3.2.2 Experimental site and data collection

*Experimental site 2018*

The field campaign in Florida, USA, was conducted in Citra (29.410N, 82.179W), at the Plant Science Research and Educa-
tion Unit (PSREU) of the University of Florida and the Institute of Food and Agricultural Sciences (UF|IFAS). Sweet corn was
planted on a sandy soil with a density of 7.9 plants m$^{-2}$, and harvested after 66 days in mid-June for human consumption. The
climate of this area in Florida is humid subtropical, and the 2018 spring growing season was characterized by high tempera-
tures, high-intensity rainfall and thunderstorms.

*Backscatter*

High temporal resolution L-band backscatter data were collected with the polarimetric University of Florida L-band Automated
Radar System (UF-LARS) throughout the growing season of 2018. This system was mounted on a Genie manlift at a height of
14 m above the ground. The scatterometer scanned the corn field with an incidence angle of 40°, and acquired 16 observations
spread throughout the day in the late season. The installation of sensors and vegetation sampling was performed outside the
arc swept by the radar. A comprehensive description of the observations and the UF-LARS system can be found in Vermunt
et al. (2020) and Nagarajan et al. (2014), respectively. *Cross-pol* is used to refer to the average of the HV- and VH-polarized
backscatter.

*Reconstruction of diurnal VWC variations for multiple consecutive days*

To support the analysis of variations in the L-band backscatter, a 10-day time series of diurnal VWC variations was recon-
structed for the 2018 data. The methodology used for the reconstruction was based on adjustments and evaluation of the sap
flow approach presented in section 3.1.1. The required sap flow and ETo data sets were similar, but slightly different. In 2018,
four sap flow sensors were installed simultaneously on four different plants and data were averaged. Gaps in the time series
were caused by disturbances in the connection with the battery or solar panel.

Meteorological data with 15-minute resolution were obtained from the nearby Florida Automated Weather Network (FAWN) weather station, located within 600 m from the experimental field. Observations of rainfall, air temperature (2m), solar radiation, relative humidity and wind speed were downloaded from the Report Generator [1]. ETo was estimated using the same Penman-Monteith approach described by Zotarelli et al. (2010).

In contrast to the 2019 data set, VWC samples were not collected to capture the full diurnal cycle. Instead, these samples were obtained four times per week. Three days at 6:00, and one of these days also at 18:00, originally to capture differences between morning and evening passes for a sun-synchronous satellite such as SMAP (Entekhabi et al. (2010)). Moreover, the presented VWC data for 2018 are averages of eight plants instead of six. The samples were used to constrain the reconstructed VWC variations.

The period of consecutive days for the analysis was limited by the availability of sap flow data. A 10-day time series was found in mid-to-late season which contained continuous sap flow and weather data, L-band backscatter, and five sampling days. On these days, samples were used to constrain the VWC record. On the five days without sampling, the VWC records were constrained either at the end of previous sampling day (forward reconstruction), or the start of next sampling day (backward). In case there was a gap between forward and backward reconstructions, the average of both was considered the best estimate of VWC.

*Soil moisture and surface canopy water*

For the analysis of sub-daily variations in the L-band backscatter, we also collected 15-minute variations in surface soil moisture, at 5 cm depth, and SCW. Together with VWC, they form the *moisture stores* of a corn field which are considered to affect sub-daily backscatter. Details of the sensors and measurements are described in section 3.1.2 and extensively in Vermunt et al. (2020).

## 4 Results

### 4.1 Seasonal and diurnal variation of vegetation water content

Fig. 2 illustrates the seasonal and diurnal variations of VWC [kg m$^{-2}$] from destructive sampling in the 2018 and 2019 campaigns. From early to mid-season, VWC increased as a result of biomass accumulation. The field corn from 2019 was allowed to senesce before harvest, resulting in a significant reduction of water storage in the plants from August 23 onward. The sweet corn from 2018 was harvested before considerable senescence.

The open markers are the non-predawn measurements, which were at 18:00 (2018) and at four evenly distributed times between sunrise and sunset (2019). The range of these latter diurnal measurements gives an indication of the amplitude of the daily cycle of VWC. On most days, the diurnal minimum was 10-20% lower than predawn water storage. An exception was July 23, on which predawn water storage was depleted by 35.4% during the day. Fig. 3 zooms in to mid-season measurements, and

---

[1]https://fawn.ifas.ufl.edu/data/reports/

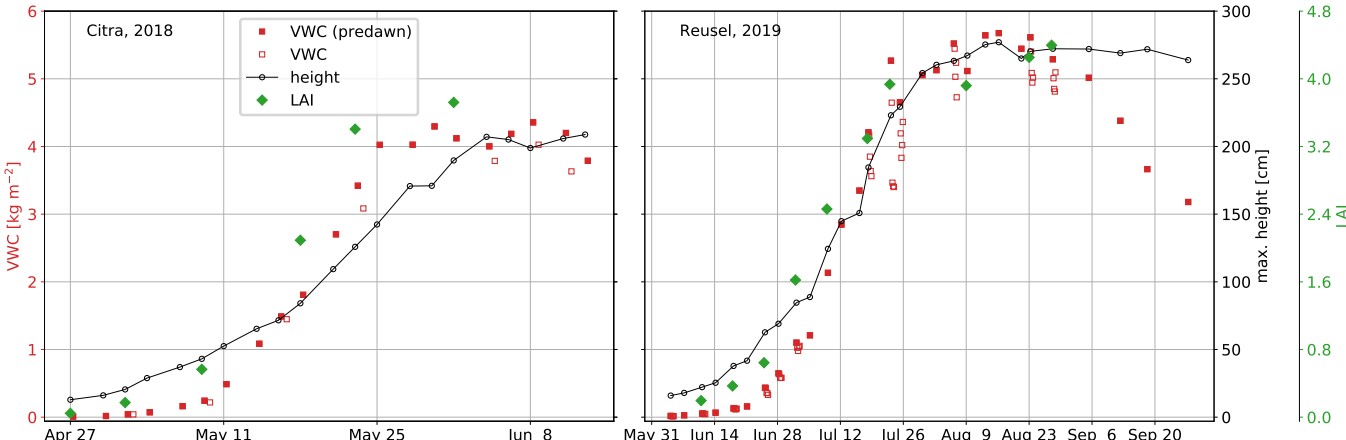

**Figure 2.** Vegetation Water Content (VWC), crop height and leaf area index (LAI) from the field experiments in Citra (2018) and Reusel (2019). Filled red markers indicate predawn measurements, while open markers indicate non-predawn measurements at 18:00 (2018) and morning to sunset (2019).

illustrates the difference between water depletion in the non-stressed conditions compared to the stressed date. The photograph was taken around the third measurement on July 23. This picture shows leaf 'rolling', a mechanism to reduce the leaf area exposed for transpiration, and a sign of drought stress. Normal-shaped leaves were observed again as a result of irrigation, which was applied right after the last sampling on July 23 in order to ensure the crop's survival.

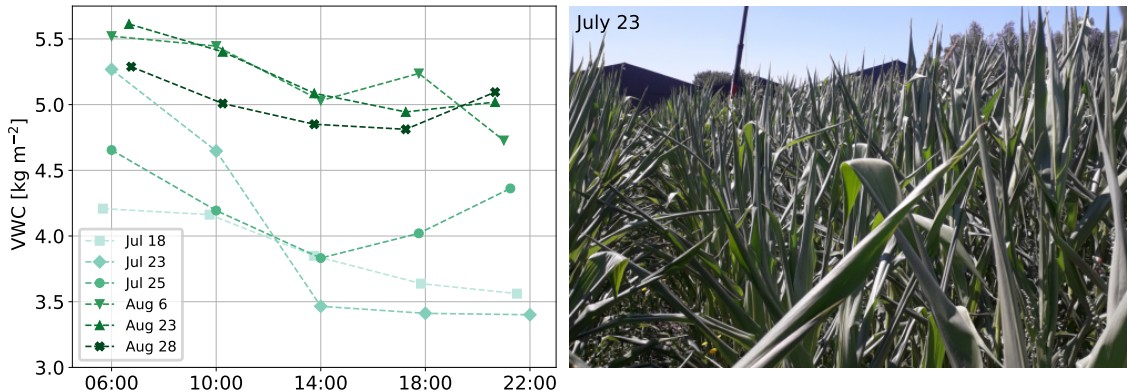

**Figure 3.** Sampled vegetation water content (VWC) in the mid-season, 2019 (left), and a picture of 'rolled' leaves (right), taken around the third measurement on July 23, as a sign of drought stress.

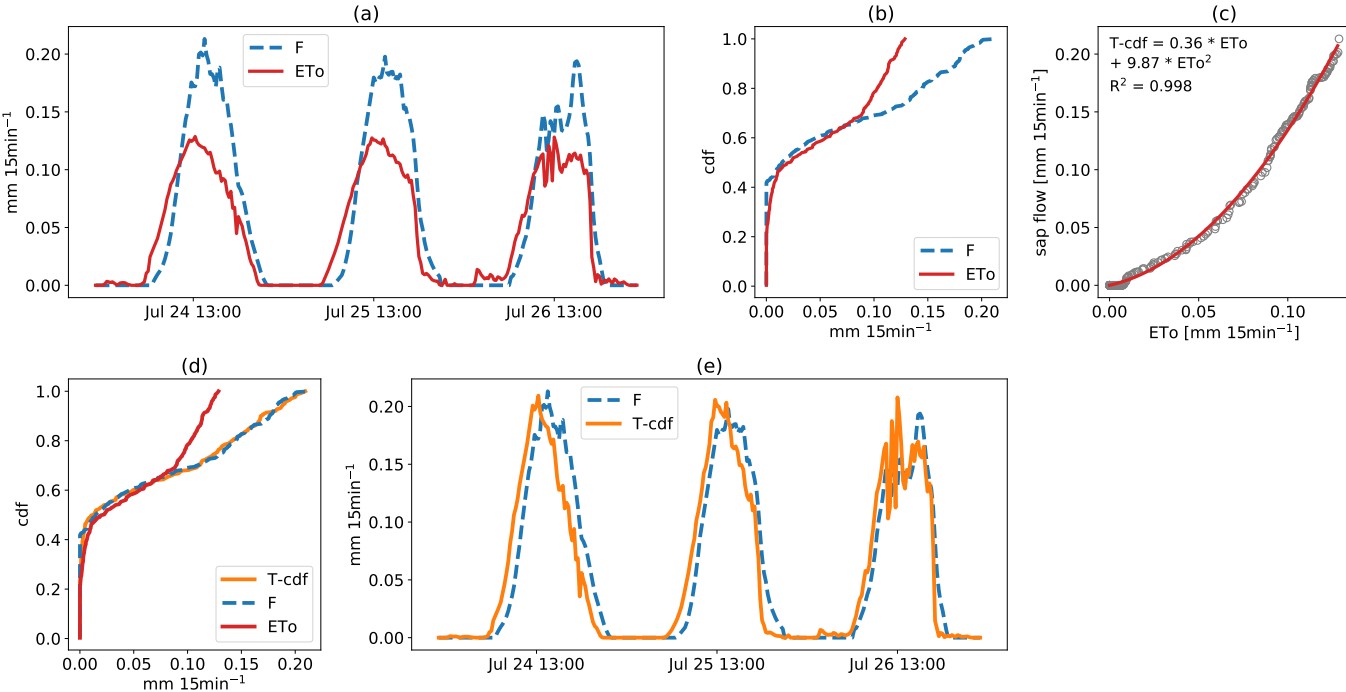

**Figure 4.** Example of ETo rescaling to approximate transpiration (2019 campaign), using the CDF-matching approach. (a) sap flow (F) and reference evapotranspiration (ETo) data from July 24-26, 2019, (b) cumulative distribution function (CDF) of both data sets in this period, (c) $2^{nd}$ order polynomial fit through ranked F and ETo data, used to derive the CDF-matched transpiration estimate (T-cdf), which was added to the CDF plot in (d). (e) shows the final result of the CDF-matching.

## 4.2 Reconstructions of continuous, sub-daily variations in vegetation water content

As described in section 3.1.1, we tested three approaches to estimate transpiration from ETo and sap flow. An alternative to the straightforward linear conversions, we proposed to test the non-linear CDF-matching principle (Table 1). Fig. 4 illustrates the procedure of estimating transpiration using this principle, using 3-days of sap flow and ETo data. We take July 25, 2019, as an example, and use the data from July 24 and 26 as well (a). On July 25, which was particularly warm and sunny, we measured a maximum temperature of 39.0 °C in the field. Fig. 4(b) illustrates the difference between the CDF's of sap flow and ETo, which is particularly evident at the 35% highest rates. At lower rates (<0.07 mm 15-minute$^{-1}$), ETo rates were slightly higher than sap flow rates. As these systematic differences between both rates may be unrealistic, a second-order polynomial was fitted through the scatter plot with ranked ETo and sap flow data (c), and was used to match the CDF's (d). The resulting T-cdf (e) was used to estimate ΔVWC at any point in time using the approach described in Fig. 5.

The procedure to reconstruct 15-minute changes in VWC is depicted in Fig. 5, again for July 25 as an example. Fig. 5(a) illustrates the effects of the three approaches to estimate transpiration from ETo and sap flow (Table 1). T-cdf and T-3d represent the CDF-matched and linear estimates of transpiration, for which 3 days of data were used: July 24-26. What stands out is that

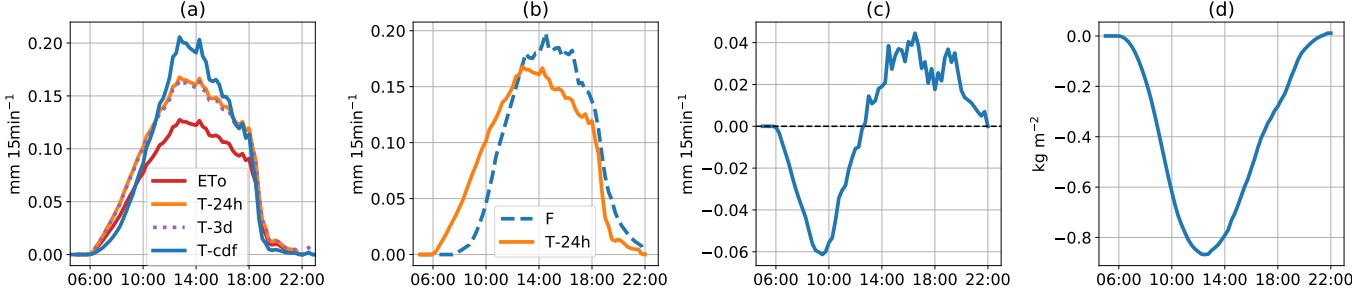

**Figure 5.** Four-step procedure to reconstruct the diurnal variation in VWC: example for July 25, 2019. Fig. (a) shows the diurnal cycles of reference evapotranspiration (ETo) and the three transpiration estimates (see Table 1), (b) shows the diurnal cycles of sap flow (F) and one of the transpiration estimates (T-24h), (c) is the difference between sap flow and transpiration, where negative values indicate depletion of water storage, and positive values indicate refill, and (d) illustrates the resulting cumulative change in stored water (ΔVWC) during the day.

the CDF-matched rescaling (T-cdf) provides a significantly higher peak, compared to the linear rescaling (T-24h and T-3d). On the other hand, when ETo rates are 0.09 mm 15min$^{-1}$ or lower, T-cdf was lower than the linear estimates. Both linear transpiration estimates were close in this particular case, which means that the ratio of the 24h sum of sap flow over ETo was close to the ratio of the 3-day sum of sap flow over ETo. Fig. 5(b) shows the diurnal cycles of basal sap flow (F) and transpiration. Here, the simplest linear transpiration estimate (T-24h) was depicted as an example. The difference between sap

flow and transpiration gave the estimated depletion and refilling of internal water storage (c). If transpiration rates exceeded sap flow rates at some point in time, the line is below zero, which indicates a depletion of water storage. Positive values indicate refilling. Finally, the cumulative difference between sap flow and transpiration represents the diurnal change in plant water storage, or ΔVWC (d). The minimum VWC was reached around 12:45, when 0.87 kg m$^{-2}$ of the predawn water storage was depleted. This is close to the maximum diurnal difference of 0.82 kg m$^{-2}$ observed on that day from destructive sampling (Fig.

3).

   Diurnal cycles of VWC were reconstructed for both linear and non-linear transpiration estimates, using ΔVWC (Fig. 5(d)) and one destructive sample (Fig. 2,3) per day as a constraint. Results were compared against the other destructive samples. The effect of both the time of the constraint, as well as the number of days considered for transpiration estimation, on the VWC reconstructions were evaluated. The RMSE's of the 2019 data are presented in tables A1-A2 in the Appendix. A general

optimal combination of time of constraint and number of days to consider could not be found. Using CDF-matched transpiration estimates resulted in a better agreement with the destructive sampling data than using linear correction in 80% of the cases. The best reconstructions from 2019 (Tables A1 and A2), are presented in Fig. 6, differentiated by the approach to estimate transpiration. Differences between environmental conditions are shown in Fig. 7. Fig. 6 illustrates the improvement of the reconstruction when using more than one day of data for the estimation of transpiration (second and third row). The upper row

clearly shows that the linear-24h approach does not allow for a difference between the start and end-of-day VWC, while the inclusion of multiple days does. Besides, the reconstruction on July 25 illustrates the possible improvement CDF-matching

can have. On July 25 and August 28, the RMSE's of the lowest plots were 8 and 12% of the amplitude of the diurnal cycles, respectively. On August 23, the agreement is poor, especially later in the day, and this percentage is 36.9%. On this day, reconstructions and samples disagree for all three approaches to estimate transpiration, but less so for the CDF matching procedure.

For the 2018 campaign, we had a maximum of two VWC samples per day. Table A3 shows the offset between one of the samples and the reconstructed VWC, which was constrained by the other sample, for June 4, 8 and 11. The lowest offsets were found when transpiration was estimated using all data (12 days), and when CDF-matching was applied. Consequently, we used the transpiration calculated based on this combination for further use of reconstructed VWC.

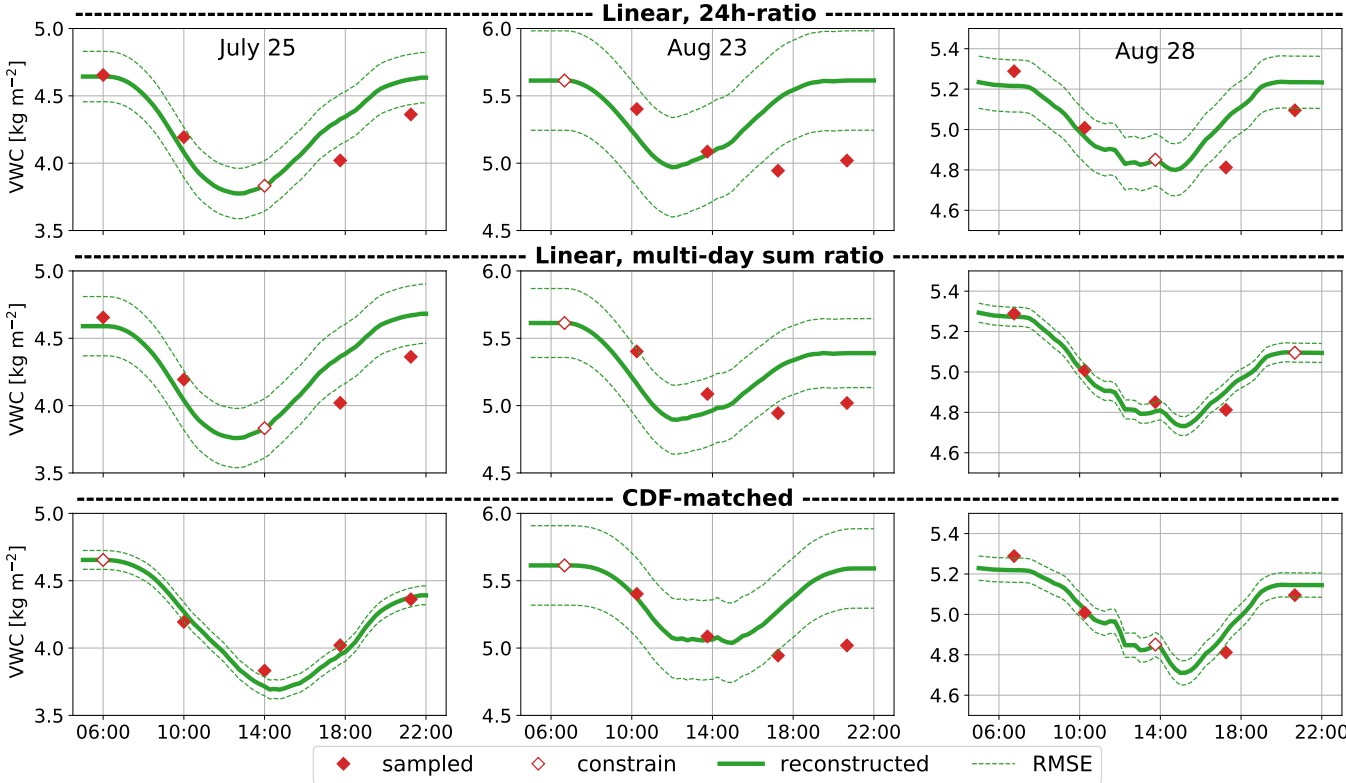

**Figure 6.** Best diurnal VWC reconstructions for July 25, August 23 and August 28 (2019) for three different methods of estimating transpiration. The upper row shows the results for using the simplest, linear estimate of transpiration. The middle row shows the reconstructions using linear estimates of transpiration, but now considering three, five and seven days rather than 24 hours. The lower row shows the results after cdf-matching, considering all data, five days and three days for the CDF-matching, respectively. The dashed green lines respresent one RMSE above and one RMSE below the reconstructed VWC. The measurement which is used to constrain the reconstructed line is accentuated with an open marker.

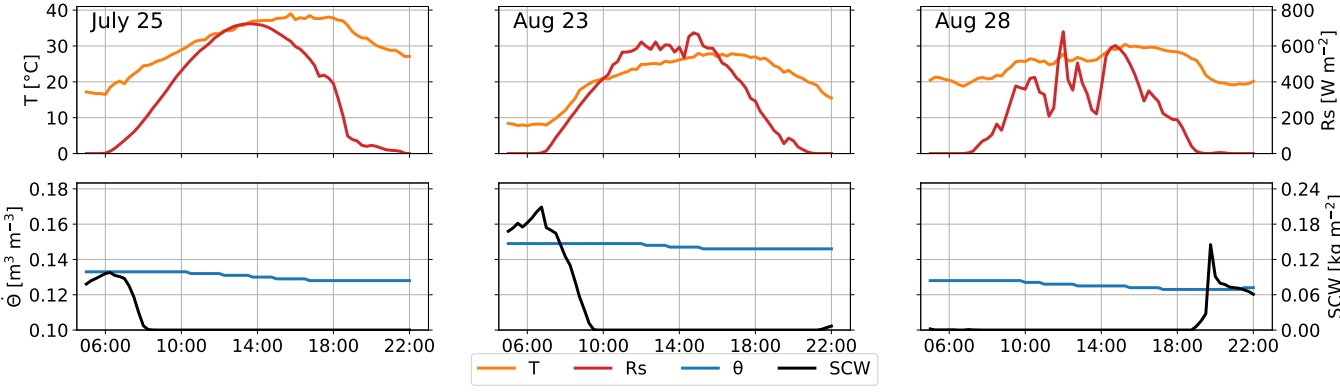

**Figure 7.** Environmental conditions on the sampling days July 25, August 23 and August 28 (2019). The upper row shows air temperature (T) and solar radiation (Rs), and the lower row shows root zone soil moisture (θ) and surface canopy water (SCW).

## 4.3 Reconstructing a record of multiple days

Fig. 8 shows the procedure for reconstructing the 10-day VWC record from 2018-data. On June 4, 8 and 11, evening samples (18:00) were used as constraints rather than predawn samples (6:00), which resulted in smaller gaps between consecutive days (Fig. 8(c)). On days without sampling, VWC records were the averages of forward or backward reconstructions. On June 9 and 10, the weighted average based on the distance to the sampling date was considered the best estimate of VWC.

The diurnal VWC pattern on June 5 and 6 seems physically implausible, because one would not expect an enormous increase of VWC on the warmest and driest day (June 5), and a drop on the most rainy/cloudy day (June 6). Despite the advantage of CDF-matching, opposed to linear conversion, to better reflect diurnal extremes, the anomalous dynamics of June 5 and 6 are not captured sufficiently.

## 4.4 The effect on sub-daily L-band backscatter

Fig. 9 illustrates the potential value of reconstructing VWC records for interpreting time series of microwave remote sensing data, in this case L-band backscatter. The upper three panels show the VV-, HH- and cross-polarized backscatter coefficients, respectively. Fig. 9(d) shows the sampled and reconstructed VWC, together with the total canopy water (CW), which is the sum of the reconstructed VWC and SCW [kg m$^{-2}$]. The latter is either rainfall interception, characterized by rapid increases and often transient because of daytime evaporation, or dew formation, which accumulates gradually during the night and dissipates quickly after sunrise. Fig. 9(e) shows the volumetric soil moisture at -5 cm depth.

Sub-daily variability of >2dB was found in all three polarizations. A sharp backscatter increase after rainfall was observed in all polarizations. Slow downward trends were also found corresponding with drydown in soil moisture. However, on a sub-daily time scale, backscatter variability shows strong similarities with diurnal patterns of canopy water (d). These diurnal cycles are most clearly visible in VV-pol. Fig. 10 zooms in to the diurnal variations for three days without rainfall: June 7, 9 and

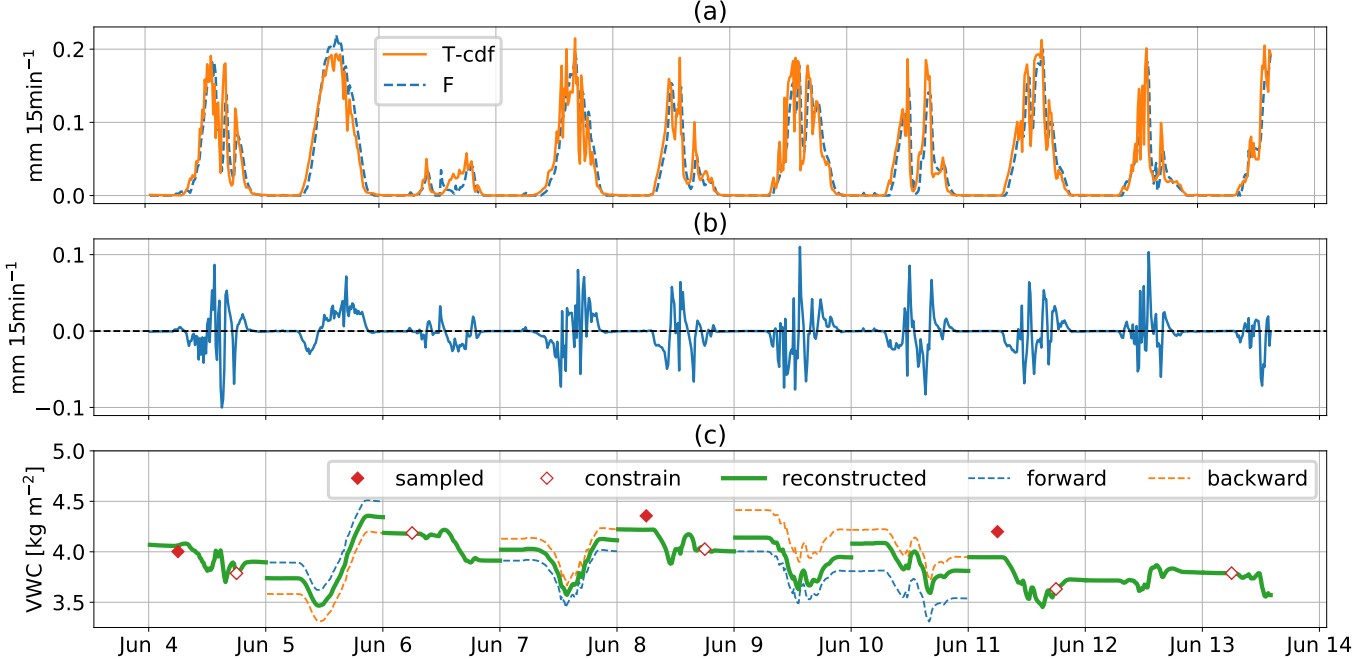

**Figure 8.** 10-day reconstruction of VWC, with (a) sap flow (F), and estimated transpiration (ETo-cdf), (b) the difference between sap flow and transpiration, and (c) the sampled and reconstructed VWC. In between sampling days, VWC estimates are the weighted average between forward and backward reconstructions from the consecutive sampling days (based on the time to the closest sampling day). The measurements which are used to constrain the reconstructed line are accentuated with open markers.

11. These days demonstrate clear similarities between the diurnal behaviour of the backscatter, mainly VV- and cross-pol, and canopy water. These similarities are particularly present in the period between midnight and mid-afternoon, when surface soil moisture is relatively stable. In fact, when randomly occurring rain events are excluded, the sub-daily backscatter behaviour can be analyzed using three distinct sub-daily periods: (1) from midnight to early morning, (2) from early morning to afternoon, (3) from afternoon to midnight. The aggregated data in Fig. 11 help to visualize the dynamics in these periods. Because rain fell more often in the afternoon and evening, the exclusion of periods with rainfall led to data aggregation across 9, 6 and 4 days in these three periods, respectively. Around midnight, dew started to form until its peak between 7:00 and 7:30, which is within an hour after sunrise around 6:30. In this same period, VWC was stable and surface soil moisture decreased slightly. VV and cross-polarized backscatter increased, following dew formation, while HH-polarized stayed relatively stable. From early morning (7:30) to afternoon (14:00), dew dissipated and VWC dropped significantly. The same holds for backscatter in all polarizations, while surface soil moisture was still relatively stable. Finally, the last period of the day is characterized by refilling of the plant's internal water storage, and a decrease of soil moisture. The fact that backscatter in all polarizations remains relatively constant between 15:00 and 19:30 suggests counterbalancing effects of soil moisture and VWC on backscatter in this period. During the last four aggregated acquisitions between 19:00 and 21:30, VV- and cross-polarized backscatter show a

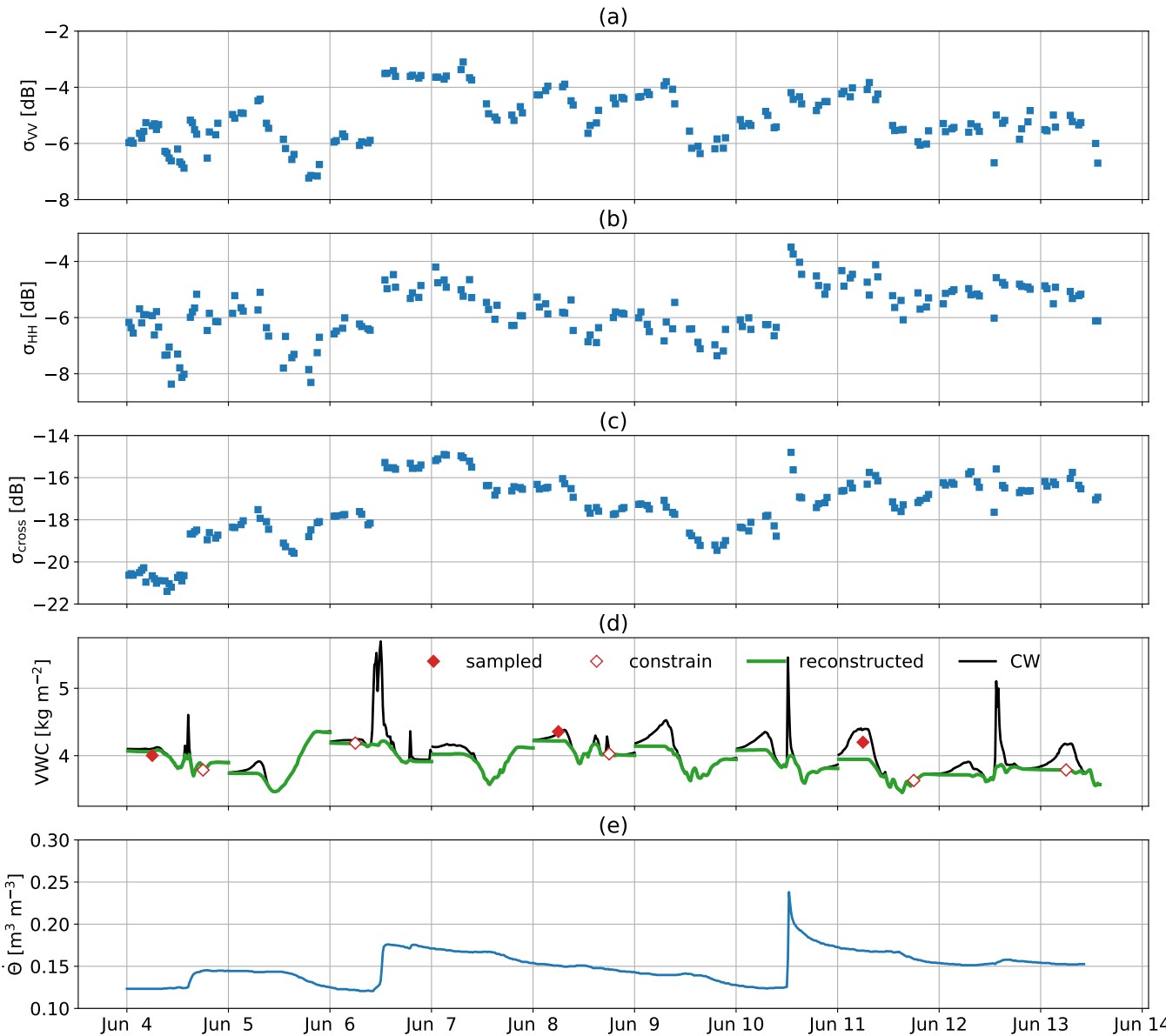

**Figure 9.** Full polarimetric L-band backscatter and separated effects for a 10-day period near the end of the growing season in 2018, with (a) VV-polarized scattering coefficient, (b) HH-polarized scattering coefficient, and (c) averaged VH and HV-polarized scattering coefficients, (d) sampled and reconstructed VWC, and total canopy water, which is the sum of reconstructed VWC and SCW, and (e) soil moisture at 5 cm depth. The measurements which are used to constrain the reconstructed line are accentuated with open markers.

slightly increasing trend similar to VWC.

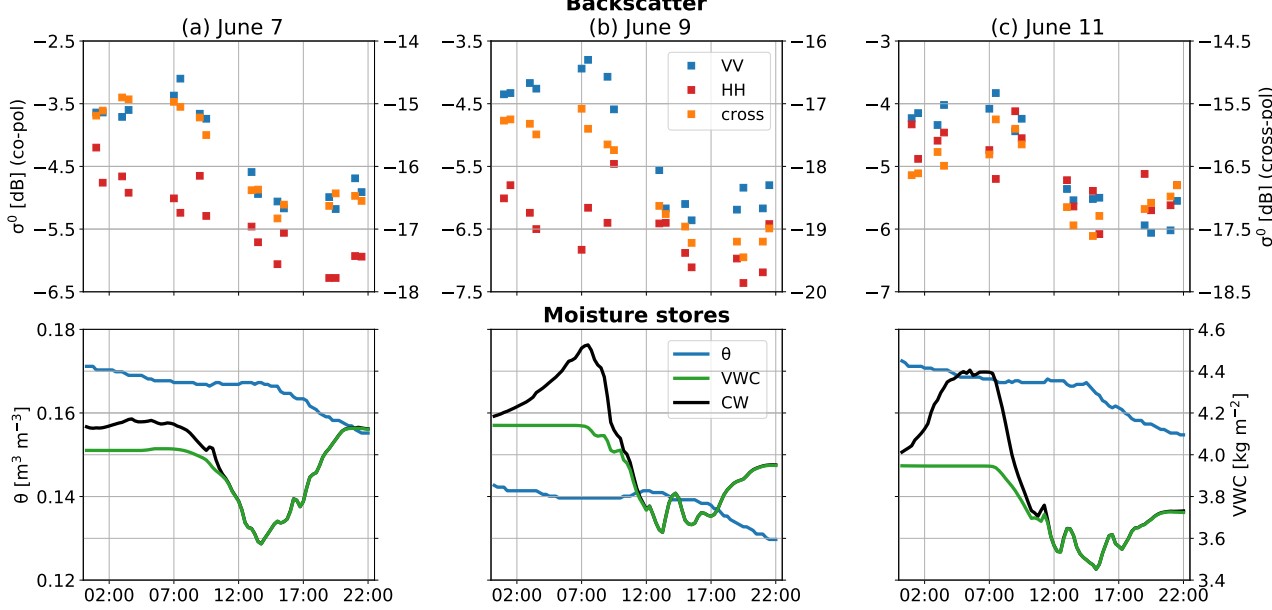

**Figure 10.** Diurnal behaviour of backscatter (VV, HH, cross-pol) and moisture (soil moisture, VWC, SCW) for three individual days without rainfall. These days were selected from the period presented in Fig. 9.

The separate effects of the different moisture stores on backscatter ($\sigma^0$) were quantified through multiple linear regression. Because we considered the VWC reconstructions from June 5 and 6 as less reliable, the period between June 7 and 13 was used for the regression. Table 2 presents the estimated regression coefficients found for this period (see equation 6). A summary of the multiple linear regression statistics is given in Table A4 in the Appendix. The regression coefficients suggest that from all

polarizations, VV-pol was most sensitive to internal vegetation water storage, and least sensitive to soil moisture. Compared to other polarizations, HH-pol was least sensitive to VWC and SCW, and most sensitive to soil moisture. Cross-pol was more sensitive to SCW than co-pols. Note that the coefficients from soil and vegetation water stores (Table 2) have non-homogeneous physical units. Nonetheless, these coefficients indicate that for a typical dry day during the campaign of 2018, e.g. June 9th, the soil moisture reduction of 0.015 $m^3 m^{-3}$ translates to a -0.4, -0.6 and -0.6 dB change in VV, HH and cross-polarized backscatter,

respectively. During the same day, VWC changed with 0.5 kg $m^{-2}$, which would translate to a change of 1.5 dB (VV), 1.2 dB (HH) and 1.2 dB (cross). This indicates that on this typical dry day, a diurnal variation in VWC leads to an almost four times higher change in VV-polarized backscatter [dB] than a diurnal change in soil moisture does. On the same day, the changes in HH- and cross-polarized backscatter [dB] were two times higher for the diurnal VWC variations than for the soil moisture drydown. The 0.4 kg $m^{-2}$ dew formation and dissipation caused $\sigma^0$ to vary with 0.2 dB (VV), 0.2 dB (HH) and 0.3 dB (cross).

Fig. 12 presents the results of using the regression coefficients (Table 2), and the time series of VWC, SCW and soil moisture, to describe diurnal variations in backscatter. Each day is constrained by the first radar observation of the day, at 01:00. Note from the $R^2$ values in Table A4 that 68-71% of the variance in backscatter is explained by the three predictors. The P-values

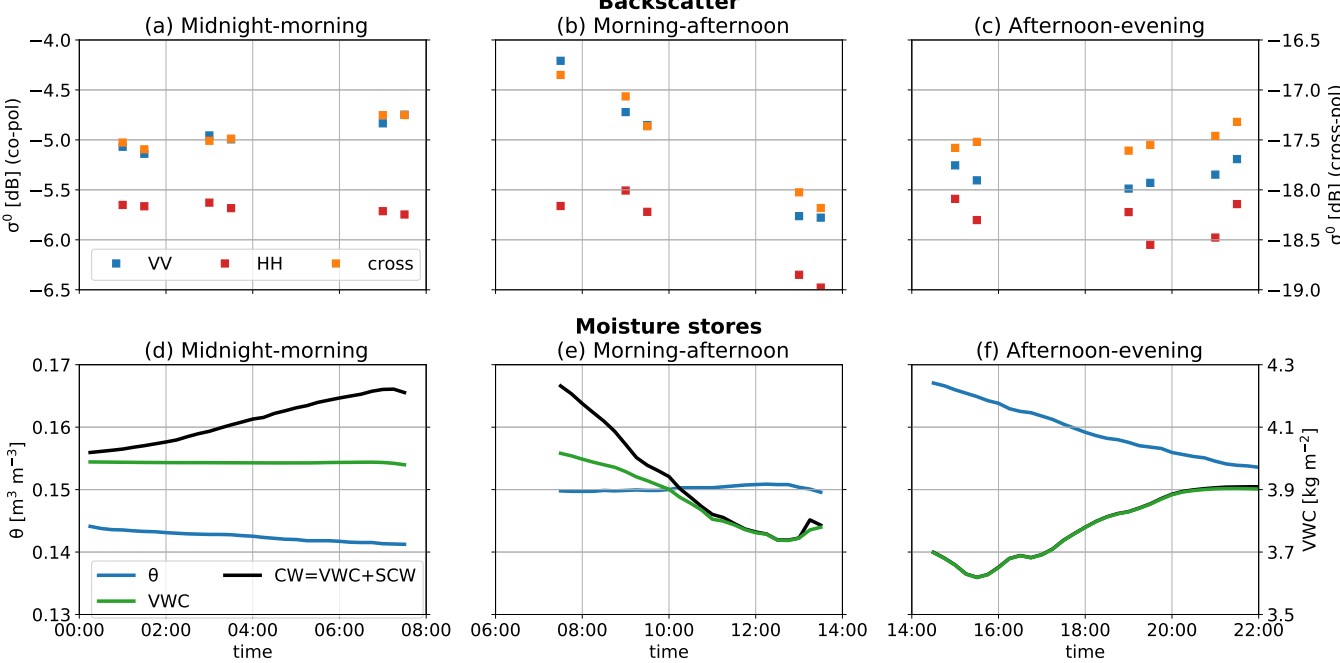

**Figure 11.** Backscatter (VV, HH, cross-pol) and moisture (VWC, CW, $\theta$) data aggregated across multiple days, and separated by part of the day: midnight-morning, morning-afternoon, and afternoon-midnight. Periods with disturbing rain events are excluded, which means that data in (a,d), (b,e), and (c,f) are aggregated across 9, 6 and 4 days, respectively. Canopy Water (CW) is SCW displayed on top of VWC.

for SCW are always higher than those for VWC and soil moisture. Nonetheless, with the exception of the SCW coefficient in the case of HH-backscatter (P> $|t|$=0.286), all P values are < 0.05, indicating statistical significance. However, note from Fig. 12(a) and (c) that the observed nocturnal backscatter increase as a result of dew formation is barely visible in the calculated backscatter. This suggests that the regression underestimates the effect of dew on backscatter.

**Table 2.** Estimated regression coefficients per polarization for the period June 7-13, 2018 (equation 6).

|  | VV-pol | HH-pol | Cross-pol |
|---|---|---|---|
| $a\,[dB/m^3m^{-3}]$ | 24.06 | 39.47 | 38.83 |
| $b\,[dB/kgm^{-2}]$ | 2.93 | 2.29 | 2.45 |
| $c\,[dB/kgm^{-2}]$ | 0.62 | 0.38 | 0.73 |

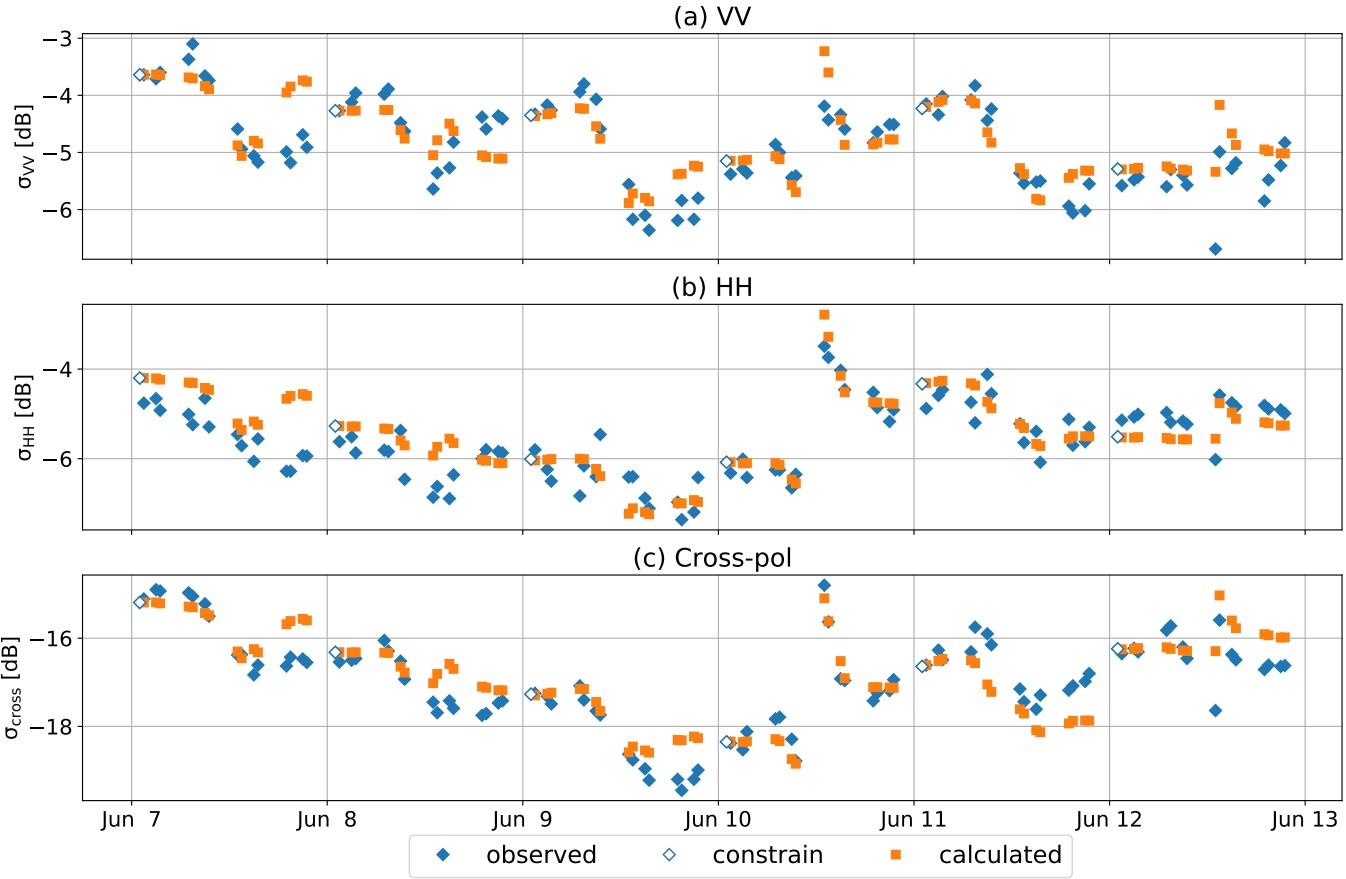

**Figure 12.** Observed and calculated (a) VV-, (b) HH-, and (c) cross-polarized $\sigma^0$ from June 7-13, 2018. The observations which are used to constrain the predictions of sub-daily $\sigma^0$-variability , $\sigma^0_{t0}$, are accentuated with open markers. Sub-daily backscatter variation is calculated using equation 6, the coefficients found by multiple linear regression (Table 2), and the time series of VWC, SCW and soil moisture.

## 5  Discussion

### 5.1  Sub-daily Vegetation Water Content estimates: observations and reconstructions

Our results showed that in non-stressed conditions, VWC depleted by 10-20 % during the day. This internal VWC withdrawal
is approximately 10-20 % of total daily transpiration, which is similar to findings from tropical and temperate broad-leaved trees (Meinzer et al. (2004); Köcher et al. (2013)). In stressed conditions, we found a 35% drop of VWC during the day.

We tested the potential of a non-destructive sap flow approach to estimate sub-daily VWC variations in corn with data from our 2019-campaign. The results confirm the possibility to estimate 15-minute variations in corn VWC with only sap flow sensors and a weather station. While the indirect estimation of transpiration could be considered a drawback of the method,
Fig. 6 has shown that the diurnal VWC cycle was represented generally well. In general, we found the best agreement between

reconstructed and sampled VWC when the daily cycle of transpiration was estimated from multi-day sap flow observations and ETo estimates. Moreover, the application of CDF-matching improved the reconstruction substantially on July 25, while on August 28, a good agreement was already found after linear correction (Fig. 6). This difference could partly be explained by the suppressing effect that dew, observed on July 25 (Fig. 7), has on transpiration (Dawson and Goldsmith (2018)), which is not captured by ETo (Langensiepen et al. (2009)). When ETo rates are low, estimated transpiration is lower after CDF-matching than after linear correction (see Fig. 4(d)). Consequently, CDF-matching mimicked the suppressing effect of dew due to the reduction in transpiration rates in the morning. When we look at the period between the peak of dew (06:00) and full dissipation (08:15) on July 25 in Fig. 6, we see that $\Delta$VWC is 0.17 kg m$^{-2}$ in the second row, while $\Delta$VWC is 0.1 kg m$^{-2}$ in the third row. This means that CDF-matching in this case led to reduction of transpiration of 0.07 kg m$^{-2}$. This is comparable to estimated dew evaporation in this period, which was 0.09 kg m$^{-2}$ (Fig. 7). The same holds for August 23, when we found a transpiration reduction of about 0.18 kg m$^{-2}$ between 6:45 and 09:45 after CDF-matching and an estimated dew evaporation of 0.20 kg m$^{-2}$ in the same period. On August 28, all dew had already dissipated before sunrise and did thus not affect transpiration. Therefore, a reduction of transpiration rates did not improve the reconstruction of VWC. These results illustrate that the suppressing effect of dew on transpiration should be taken into account when one estimates transpiration with a weather station or flux tower.

Another effect of CDF-matching was that the highest ETo rates resulted in higher estimates of transpiration compared to those obtained using linear corrections (see Fig. 4(d)). This was particularly apparent under sunny conditions such as on July 25 and August 23. This means that transpiration rates exceeded sap flow rates for a longer period. Together with the gradual depletion of internal VWC in the morning, this led to a much better agreement, and a shift of a diurnal minimum towards the afternoon. However, the poor agreement between sampled and reconstructed VWC in the evening of August 23 could not be explained by extreme hydrometeorological conditions, growth stage or drought stress. Other potential contributors to the poor agreement could be unaccounted for errors in the sap flow, weather data or samples. The cloudier conditions on August 28 (Fig. 7) could explain the small difference between linear corrections and CDF-matching.

When the methodology with CDF-matching was applied to the 10-day period from our 2018 campaign, the diurnal minima of reconstructed VWC matched excellently with the diurnal minima in the backscatter in most cases (Fig. 9). This could be explained by the daily dew formation and high temperatures in this period. However, discontinuities were observed between consecutive days (Fig. 8), which might be related to the temporal resolution of the observations and the estimation of transpiration fluxes. The temporal resolution of the sensor observations was 15 minutes. At the same time, we found phase differences between ETo and sap flow in the order of 15-45 min, which was consistent with previous studies on corn (e.g. Langensiepen et al. (2009)). Increasing the ratio between phase difference and observation resolution would increase the robustness of the method. A potential solution would therefore be to increase the temporal resolution of the sensor observations. Another potential solution is related to the estimation of transpiration fluxes. Ideally, a flux tower would be used for ET estimates through the eddy covariance method, as it is a more direct measurement and widely considered as the most accurate technique for ET measurements at field scale (Zhang et al. (2014); Maltese et al. (2018); Oguntunde et al. (2004)). Improved ET estimates may also reduce or eliminate the need to include CDF-matching. As direct ET measurements also include evaporation from

SCW and soil, it is advised to include leaf wetness sensors and micro-lysimeters (Ding et al. (2013)) to provide quantitative estimates of evaporation and determine transpiration from ET measurements. Including several *in situ* sensors of each type (e.g. leaf wetness, sap flow etc.) ensures that the quantities capture field-scale dynamics.

## 5.2 Interpreting sub-daily backscatter

In Vermunt et al. (2020), sub-daily L-band backscatter variations were attributed to VWC, SCW and soil moisture. However, the lack of sub-daily VWC data points complicated quantifying the relation between backscatter and the individual moisture stores. The VWC records generated in the current study allowed us to understand sub-daily backscatter variations with unprecedented detail, and to describe the relative backscatter sensitivity to the different moisture stores.

The results presented here indicate that the effects of sub-daily variations in VWC on backscatter are considerable. Our regression analysis suggested that, on a typical dry day, the diurnal cycle of VWC led to a two (HH- and cross-pol) to almost four (VV-pol) times higher change in backscatter than the soil moisture drydown did. Note that these ratios can be different when either VWC or soil moisture content substantially change (Brisco et al. (1990)), or when crop structure changes during the day (Kimes and Kirchner (1983)). Backscatter sensitivity to VWC dynamics were most clearly observed in the period between sunrise and mid-afternoon, when both dropped significantly. During mid-afternoon to sunset, we observed a constant to slightly increasing VV- and cross-polarized backscatter, which illustrated the opposite effects of VWC refilling and soil moisture drop on backscatter. Nocturnal backscatter dynamics demonstrated the sensitivity of VV- and cross-pol to SCW.

In general, our results showed that VV-pol was more sensitive to variations in VWC than HH-pol, and less sensitive to variations in soil moisture. This is in agreement with Joseph et al. (2010), who described a larger attenuation of the soil return by vegetation for VV-pol compared to HH-pol in a study on the L-band backscattering of corn. An explanation for this difference was given by Stamenković et al. (2015), who described that at VV and HV polarizations, vertical corn stems attenuate the double-bounce scattering at L-band, which results in lower contribution from the soil. As a consequence, volume scattering and the corresponding contribution from vegetation becomes dominant. At HH-pol, there is less attenuation of the double bounce effect, which explains a higher sensitivity to soil moisture (Table 2).

Moreover, the nocturnal VV- and cross-polarized backscatter increase in Fig. 9 and 11 could be attributed to dew formation only, because VWC was stable during the night and soil moisture was constant or slightly decreased. Stable nocturnal VWC can be expected for crops with a hydraulic capacitance similar to or lower than corn, and sufficient soil moisture availability. For vegetation with a larger hydraulic capacitance or low soil moisture availability, nocturnal refilling of VWC could be expected (Maltese et al. (2018)), which could complicate the separation of signals from VWC and SCW.

Fig. 9-11 and Table 2 showed that, compared to HH-pol, VV- and cross-polarized backscatter were not only more sensitive to changes in VWC, but also to changing SCW. This is in agreement with previous findings from Brancato et al. (2017), who found a stronger effect of SCW on S- and C-band differential interferometric observables in VV polarization compared to other polarizations, particularly for vertically oriented crops as corn. This could be related to increased scattering from wet leaves in combination with the dominance of volume scattering in VV and cross polarizations. However, it seems that the SCW coefficients (c) for VV and cross-pol in Table 2 underestimate the effect of dew on backscatter, as the nocturnal increases in

calculated $\sigma^0_{VV}$ and $\sigma^0_{cross}$ in Fig. 12 are lower than observed. This could partly be addressed by improved SCW estimates, for example through inclusion of more leaf wetness sensors distributed in the canopy (Vermunt et al. (2020)). Moreover, additional research is needed to provide more insight into the scattering mechanisms under the presence of SCW, for example by considering SCW in physical backscattering models.

## 6  Conclusions

The potential of using radar for (eco)hydrological studies is limited by the challenge to separate signals from soil and vegetation on a sub-daily timescale. To gain better understanding of what controls sub-daily backscatter behaviour, we analyzed tower-based polarimetric L-band observations from a corn field using unique estimates of moisture fluctuations in vegetation and soil.

A method developed by the tree physiology community was adapted to estimate continuous variations in corn plant water
content with unprecedented detail. The adaptations were related to the estimation of transpiration. The best agreement between sampled and estimated VWC was found when transpiration estimates were obtained after the removal of systematic differences between ETo and sap flow. In non-stressed conditions, predawn VWC decreased by 10-20% during the day.

Complementing the resulting record of VWC with records of soil moisture and previously estimated surface canopy water allowed us to interpret the sub-daily behaviour of polarimetric L-band observations. The results showed a significant effect of
460 diurnal VWC cycles on L-band backscatter when the plants reached their maximum size. The highest and lowest sensitivity to VWC was found in VV- and HH-polarized backscatter, respectively. The regression results suggested that the backscatter behaviour on a typical dry day was two (HH, cross-pol) to four (VV) times more determined by the VWC cycle than by soil moisture. Nighttime increases in VV- and cross-polarized backscatter were a result of dew formation only.

The results presented here provide unique insight into the potentially confounding influence of surface and internal vegeta-
465 tion water content variations on backscatter, particularly in the interpretation of sub-daily radar observations. These findings are directly relevant for current and upcoming L-band missions, but also for the design of future spaceborne SAR missions for land applications. In particular, this study highlights the potential difference in relative importance of VWC, SCW or soil moisture dynamics depending on the overpass time. This is particularly relevant given the imminent availability of sub-daily observations from e.g. the Iceye and CapellaSpace constellations.

As radar observations are increasingly used to study plant water status, the presented sap flow method is a promising way to validate sub-daily satellite observations with just meteorological data and sap flow sensors, without laborious sub-daily destructive sampling. The method is expected to be most robust when the temporal resolution of the sap flow and ET observations are significantly smaller than the phase difference between the two, which depends on the species. The number of sensors required to capture VWC variations at footprint scale is expected to depend on the footprint size, and the spatial heterogeneity

of vegetation type and factors influencing moisture supply and demand. Potentially, global database networks for sap flow measurements, i.e. *Sapfluxnet* [2], and flux tower measurements, e.g. *Fluxnet* [3] and *Ameriflux* [4] can play an important role here.

Moreover, the utility of the tested sap flow method goes well beyond radar remote sensing. It also has huge potential for validating and interpreting a wide range of other remotely sensing techniques that are sensitive to vegetation water, such as passive microwave remote sensing, Global Navigation Satellite Systems (GNSS) and Cosmic Ray Neutron Sensors.

---

[2]http://sapfluxnet.creaf.cat
[3]https://fluxnet.org/
[4]https://ameriflux.lbl.gov/

 **Appendix A**

**Table A1.** Root mean squared error (RMSE) between reconstructed and sampled VWC. The rows represent time of constraining the recon-struction, while the columns represent the considered period for linear ETo correction

|  | July 25 | | | August 23 | | | | | August 28 | | | | |
| --- | --- | --- | --- | --- | --- | --- | --- | --- | --- | --- | --- | --- | --- |
|  | 1 day | 3 days | all data | 1 day | 3 days | 5 days | 7 days | all data | 1 day | 3 days | 5 days | 7 days | all data |
| predawn | 0.212 | 0.272 | 1.107 | 0.369 | 0.310 | 0.256 | 0.282 | 0.547 | 0.178 | 0.097 | 0.095 | 0.063 | 0.352 |
| morning | 0.314 | 0.369 | 1.082 | 0.500 | 0.444 | 0.389 | 0.416 | 0.655 | 0.176 | 0.110 | 0.108 | 0.078 | 0.315 |
| afternoon | 0.187 | 0.220 | 0.704 | 0.375 | 0.346 | 0.318 | 0.331 | 0.468 | 0.129 | 0.090 | 0.089 | 0.075 | 0.227 |
| evening | 0.266 | 0.321 | 1.036 | 0.446 | 0.392 | 0.337 | 0.364 | 0.601 | 0.206 | 0.138 | 0.136 | 0.106 | 0.351 |
| sunset | 0.247 | 0.311 | 1.131 | 0.516 | 0.448 | 0.379 | 0.413 | 0.706 | 0.150 | 0.074 | 0.072 | 0.047 | 0.317 |

**Table A2.** Root mean squared error (RMSE) between reconstructed and sampled VWC. The rows represent time of constraining the recon-struction, while the columns represent the considered period for CDF-matching

|  | July 25 | | | August 23 | | | | | August 28 | | | | |
| --- | --- | --- | --- | --- | --- | --- | --- | --- | --- | --- | --- | --- | --- |
|  | 1 day | 3 days | all data | 1 day | 3 days | 5 days | 7 days | all data | 1 day | 3 days | 5 days | 7 days | all data |
| predawn | 0.155 | 0.140 | 0.070 | 0.303 | 0.380 | 0.295 | 0.310 | 0.458 | 0.135 | 0.112 | 0.153 | 0.149 | 0.379 |
| morning | 0.114 | 0.104 | 0.124 | 0.296 | 0.390 | 0.313 | 0.331 | 0.508 | 0.121 | 0.078 | 0.100 | 0.088 | 0.286 |
| afternoon | 0.140 | 0.136 | 0.125 | 0.309 | 0.351 | 0.311 | 0.319 | 0.402 | 0.091 | 0.060 | 0.075 | 0.067 | 0.212 |
| evening | 0.094 | 0.081 | 0.113 | 0.244 | 0.333 | 0.259 | 0.276 | 0.451 | 0.142 | 0.084 | 0.103 | 0.083 | 0.306 |
| sunset | 0.177 | 0.162 | 0.083 | 0.471 | 0.548 | 0.460 | 0.474 | 0.623 | 0.102 | 0.070 | 0.106 | 0.097 | 0.325 |

**Table A3.** Offset between reconstructed and sampled VWC. The rows represent the method used for transpiration estimation, while the columns represent the considered period.

|  | June 4 | | | | June 8 | | | | | June 11 | | | |
| --- | --- | --- | --- | --- | --- | --- | --- | --- | --- | --- | --- | --- | --- |
|  | 1 day | 3 days | 5 days | all data | 1 day | 3 days | 5 days | 7 days | all data | 1 day | 3 days | 5 days | all data |
| linear | 0.202 | 0.250 | 0.149 | 0.055 | 0.412 | 0.071 | 0.241 | 0.022 | 0.022 | 0.556 | 0.790 | 0.739 | 0.543 |
| cdf | 0.134 | 0.180 | 0.185 | 0.063 | 0.292 | 0.106 | 0.209 | 0.147 | 0.128 | 0.456 | 0.476 | 0.521 | 0.267 |

**Table A4.** Summary of multiple linear regression results.

| | VV-pol | | | HH-pol | | | cross-pol | | |
|---|---|---|---|---|---|---|---|---|---|
| | θ | VWC | SCW | θ | VWC | SCW | θ | VWC | SCW |
| coeff. | 24.0643 | 2.9340 | 0.6190 | 39.4680 | 2.2879 | 0.3759 | 38.8273 | 2.4463 | 0.7293 |
| std. err | 2.600 | 0.262 | 0.302 | 3.019 | 0.304 | 0.350 | 2.906 | 0.293 | 0.337 |
| t | 9.254 | 11.203 | 2.051 | 13.075 | 7.526 | 1.073 | 13.363 | 8.360 | 2.163 |
| P > \|t\| | 0.000 | 0.000 | 0.043 | 0.000 | 0.000 | 0.286 | 0.000 | 0.000 | 0.033 |
| [0.025 | 18.900 | 2.414 | 0.020 | 33.474 | 1.684 | -0.320 | 33.058 | 1.865 | 0.060 |
| 0.975] | 29.228 | 3.454 | 1.218 | 45.462 | 2.892 | 1.072 | 44.597 | 3.027 | 1.399 |
| $R^2$ | | 0.686 | | | 0.690 | | | 0.715 | |
| Adj. $R^2$ | | 0.675 | | | 0.680 | | | 0.706 | |

*Author contributions.* PV, SSD and NvdG were responsible for the conceptualization, methodology, formal analysis, investigation, visualization and writing (original draft preparation). JJ provided resources (scatterometer data). PV and SK conducted the ground-measurements. SSD and NvdG provided supervision. All authors contributed to writing (review and editing).

*Competing interests.* The authors declare that they have no conflict of interest.

*Acknowledgements.* This project was supported by Vidi Grant 14126 from the Dutch Technology Foundation STW, which is part of The Netherlands Organisation for Scientific Research (NWO), and which is partly funded by the Ministry of Economic Affairs. The experiment in 2018 was made possible by infrastructural and technical support from the Agr. and Biol. Eng. Dept. and PSREU at the University of Florida. The authors wish to acknowledge the help from Daniel Preston, Patrick Rush, Eduardo Carrascal, and James Boyer and his team in particular. The experiment in 2019 was made possible by infrastructural and technical support from Jacob van den Borne, Paul van Zoggel and their team. Vineet Kumar participated in the collection of field data in 2019.

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
