# Peer review of "Extrapolating Continuous Vegetation Water Content To Understand Sub-daily Backscatter Variations"

_Hydrology and Earth System Sciences, 2021_

## Author Comment (AC1)

Reviewer comments in black
Reply to comments in blue

Vermunt et al. use non-destructive sap flow measurements to estimate the diurnal cycle of vegetation water content and then relate it to microwave radar backscatter. This paper is of high relevance to ongoing microwave vegetation measurements and answering large-scale ecosystems questions. I am in support of this work given the low amount of ground measurements and available techniques and consequently high uncertainty in microwave vegetation retrievals. It creatively uses a known application in plant physiology and ecohydrology for microwave remote sensing validation. I think the study is well done and is a great contribution. I ask that the authors consider some comments here before publication.

I do not wish to remain anonymous. -Andrew Feldman

Response: Thanks to Andrew Feldman for the careful consideration of the manuscript and the constructive comments. Below we have addressed the comments in blue. The line numbers in our replies refer to the revised manuscript.

**Major comment**

I think the methodology needs a clearer section or paragraph that explicitly outlines the method used here, its advantages and disadvantages, assumptions, and how the method can be modified in scenarios of different vegetation types (tree instead of corn). This could be a modification of Section 2. The sections afterward can expand on this as they currently do with section 3 and onward. While reading the methods, I felt as if I was finding out more components required for the method as it went along. It also seems like some steps are optional or can change for different types of vegetation (see my comments below). Be clearer earlier that sap flow sensors, destructive sampling, and weather stations are needed and that this approach is somewhat specific for corn or other herbaceous vegetation types. Lines 54-55 motivate the method as a standard approach used in previous studies, but this method seems different because a sap flow sensor could not be placed in the crown and transpiration needs to be modeled. Furthermore, destructive sampling needs to be used to constrain the VWC estimates (though I am not sure this is always needed; see below). If a different vegetation type other than corn is used, the method can become more reliable because one can use two sap flow sensors and not have to model transpiration (other than relying on an additional assumption about small leaf capacitance). Since this is in part a methods paper, a more organized overview of the method can make the method more reproduceable or easier to modify.

Response: Thanks for the suggestions. We have reorganized the methods section based on your suggestions and those from the other two reviewers. We have structured the new Data and Methods section by separating the two objectives (section 3.1 and 3.2), and their corresponding data description. The adjustments of the methodology and corresponding data requirements are now grouped together and described before the details about data collection. We made a clearer distinction between a description of the original method applied to trees (section 2) and the specific modifications, assumptions and data requirements when this method is applied to corn (section 3.1). Please find answers on specific comments below.

**Line specific comments**

(1) Line 21: Here or further down, an explicit definition of how vegetation water content is traditionally defined is needed. "Water content" can be confusing because it could be a total water volume (as is the case traditionally with VWC) or could mean a ratio to the dry or total volume (as for soil moisture or soil water content). Therefore, a definition of kg/square meter or other used here would be helpful.

Response: Agreed. We added a definition of VWC in lines 21-22.

(2) Line 47-49: This is an excellent introduction. The main thing I feel that is missing is I am wondering if the authors could be more descriptive here of the other VWC in-situ sample options, how prevalent they are, and why they didn't choose them. A few things I am wondering: is the destructive sampling method the most common for radar validation? What specific destructive methods are used (oven drying leaves, branches, etc.)? Why not measure leaf/stem water potentials using automated psychrometers (Guo et al., 2019) since those sensors can provide rapid measurements (then mention why that does not directly provide water volume)? Have others used psychrometers or water potential measurements for radar validation? Another approach used in radiometry for VOD was to use water potential measurements and biomass to estimate VWC (Momen et al., 2017). Similarly, VOD was related to diurnal variations of leaf water potentials (Holtzman et al., 2021). I don't think the authors need to provide a large description of this (or cite these papers for that matter). I think it may provide more context and perhaps strengthen the motivation to choose the sap flow method by contrasting with other known options.

Response: We acknowledge that the suggested automated psychrometer and water potential measurements are very interesting, and potentially useful to add to the instrumentation in the field. Although water potential is the most useful descriptor for water status and from the plant hydraulic perspective, it does not have a one-on-one relation with dielectric constant and thus backscatter. The best possible measurements for validation of radar observations are therefore direct measurements of plant water content rather than water potential. Indeed, destructive sampling is the most direct and common method for measuring VWC for validation in agricultural terrains. For woody constituents in trees, dendrometers have been used to infer water content non-destructively after detrending, and similarly, reflectometry (TDR and FDR) and capacitance-style sensors have been used to derive water content indirectly by measuring dielectric permittivity (Konings et al. (2021)). However, as far as we know, these sensors were never used for non-woody tissue. We have addressed this in lines 54-58, which now read: "For woody constituents in trees, dendrometers have been used to infer water content non-destructively after detrending, and similarly, reflectometry (TDR and FDR) and capacitance-style sensors have been used to derive water content indirectly by measuring dielectric permittivity (Konings et al. (2021)). Moreover, a water balance-style approach using sap flow sensors have been used by the tree physiology community to estimate diurnal changes in tree stem water storage (Goldstein et al. (1998); Meinzer et al. (2004); Cermak et al. (2007); Phillips et al. (2008); Köcher et al. (2013))."

(3) Line 52: A more specific research question/objective could be helpful here. This objective has been broadly pursued before. The authors are specifically testing whether a non-destructive sap flow technique can measure VWC and thus be used to validate radar diurnal VWC measurements, which is a great endeavor that should be explicitly stated.

Response: Lines 52-58 of the original manuscript were rewritten to make the objectives and used methodology more explicit, and now read: "The objectives of this study were to test the potential of a non-destructive sap flow technique for estimating sub-daily VWC variations in a herbaceous plant, and to use these estimates to better understand what controls sub-daily variations of L-band backscatter. Specifically, we adapted a

methodology developed by the tree physiology community, described in section 2, to estimate 15-minute changes in corn VWC using sap flow sensors and a weather station. An extensive data set from a field campaign in the Netherlands in 2019 was used to evaluate the adapted method against diurnal cycles of VWC obtained by destructive sampling. Finally, the technique was applied to reconstruct sub-daily VWC variability of multiple consecutive days from another field campaign in Florida in 2018. In this campaign, high temporal resolution tower-based polarimetric L-band backscatter was collected. The reconstructed VWC was used, together with simultaneously collected soil moisture, surface canopy water (SCW), to gain better understanding of what controls sub-daily backscatter behaviour."

(4) Lines 71-78: I became confused here because I thought in line 71 that this approach is applied here. Then I found out that it wasn't in line 79. Please only mention the assumptions applicable here to a single sensor and estimated transpiration. Then give more detail about the assumptions. You could argue that this approach circumvents the first assumption which could be flawed; the first assumption I think suggests that capacitance is negligible in the leaves and is larger lower in the canopy (trunks and lower parts of branches). This may not always be true for succulents and large trees. With the second assumption and full approach here, I wonder whether day to day variations can still be measured with this approach if a storage term is estimated and stem flow measurements are consistent.

Response: In the revised manuscript, we have made it clearer that the mentioned lines relate to previous research on trees, by (1) changing the subtitle of section 2 to 'Estimating diurnal variations in tree water content using sap flow probes', (2) adding 'In these studies on trees, ...' to the original line 71, and (3) moving the last paragraph of this section, which was related to the adaptations we made for a corn crop, to the reorganized methodology section. We think that mentioning the assumptions made by tree physiologists in section 2 is crucial to understand the adjustments required to apply the methodology to corn plants (section 3.1.1). We hope that the re-organization of the methodology section makes this clearer.

(5) Line 109-113: This paragraph appears to give some extraneous information. It might be helpful to only mention the measurements relevant to this study. The authors are not trying to minimize day-to-day weather variations here.

Response: In response to this comment, we removed the 'minimize day-to-day weather variations' part. However, since it can be confusing that we used two data sets, we think it is still insightful to explain why the VWC data sets were different in the two campaigns. The reason that we used the 2019 VWC data set for evaluating the sap flow technique, was because we did not collect full diurnal cycles of destructively sampled VWC in 2018. The original lines 109-113 are adjusted and now read: "In contrast to the 2019 data set, VWC samples were not collected to capture the full diurnal cycle. Instead, these samples were obtained four times per week. Three days at 6:00, and one of these days also at 18:00, originally to capture differences between morning and evening passes for a sun-synchronous satellite such as SMAP (Entekhabi et al. (2010)). Moreover, the presented VWC data for 2018 are averages of eight plants instead of six. The samples were used to constrain the reconstructed VWC variations." The new line numbers are 234-240.

(6) Line 118: For clarity, one sensor is placed at the base of the plant for each plant (as suggested by lines 79-86)?

Response: Please find in lines 119-120 (original manuscript) that the sensors are 'wrapped around a corn stem, about 20 cm above the ground, ...'. Added 'near the base of the stem' for clarity to original line 118 (now line 154), which now reads "Sap flow was monitored near the base of the stem using stem-flow gauges produced by Dynamax Inc. (Houston, TX, USA)." .

(7) Line 144-146: Consider showing an equation of this here.

Response: An equation is now shown in Table 1.

(8) Line 169: What time of day are these samples from?

Response: Added 'spread throughout the day' to the sentence. Radar acquisition times are shown in the table below.

Table 1: Radar acquisition times (EDT)

| | | | |
|---|---|---|---|
| 01:00 | 07:00 | 13:00 | 19:00 |
| 01:30 | 07:30 | 13:30 | 19:30 |
| 03:00 | 09:00 | 15:00 | 21:00 |
| 03:30 | 09:30 | 15:30 | 21:30 |

(9) Line 180: Since modeling transpiration can be viewed as the largest uncertainty of the method, I would add more details about how P-M equation was used and choices made for certain parameters (like roughness height and others).

Response: We estimated transpiration from reference evapotranspiration (ETo) and sap flow data. ETo was calculated based on the sequence of equations presented in Zotarelli et al. (2010). ETo is the ET from a hypothetical, optimally growing grass reference crop, and was introduced to study the evaporative demand of the atmosphere independent of crop characteristics and management practices. Site-specific inputs are (1) weather station data (air T, solar radiation, wind speed), (2) field location (latitude, elevation above sea level), and (3) Day of Year. Hence, for the calculation of ETo, we did not have to make choices for crop-specific parameters like roughness height etc.

(10) Line 184: Using the data to constrain and validate here becomes somewhat circular. I think the method is generally fine. However, I would note that I don't think this step is entirely necessary – I think the authors can simply try to compare the temporal dynamics of the reconstructed VWC and measured VWC and not worry about correcting the bias too much.

Response: Indeed, our main objective here was to compare the temporal dynamics of the reconstructed VWC and measured VWC. We agree that for this purpose, we wouldn't need to calculate the RMSE for the five different samples. However, another sub-question we had was 'at which time of day should we sample if we would want to have the most realistic diurnal cycle of VWC?' To address this question, we needed to calculate the RMSE between samples and reconstructions, using all five samples once to constrain the reconstruction.

(11) Line 200: Is it true that the VWCt0 reference is needed to get the day to day dynamics right while the

VWCt (= sap flux – transpiration) term is all that is needed to explain the backscatter diurnal variations within a day? If so, I would be more explicit about this. This can be seen where, in eq. 3, the constant VWCt0 term would mostly get lumped into the y intercept term. VWCt0 are mostly a magnitude scaling and won't change the relationship between the VWCt (= sap flux – transpiration) term and backscatter within a day. The VWCt0 is essentially picking up on the biomass and total water storage changes day to day. VWCt is effectively the storage anomaly which is all that is needed to evaluate the backscatter anomaly.

The consequence is that if one is only interested in the subdaily variations, the destructively sampled VWCt0 reference used to scale the VWC is not necessarily needed and is an extraneous step (this can be seen with using a panel regression in place of eq. 3 where the eq. 3 regression is effectively applied separately to each day's diurnal variations). If true, I think this idea should be mentioned. Perhaps the extra step to use destructive VWC sampling each day is to evaluate day to day changes in VWC. The point is that I think one can test the time dynamics of backscatter at large spatial scales using only sap flux and transpiration estimation without needing labor intensive, destructive methods to constrain the magnitude of VWC. If I am wrong, consider clarifying the issue in the text.

Response: Indeed, VWCt0 won't change the relationship between VWCt and backscatter within a day. VWCt0 is indeed used for magnitude scaling. If we would not use VWCt0, the reconstructed diurnal variations would be the same. In equation 6 (former eq. 3), VWCt0 is not the destructive sample, but simply the reconstructed VWC at t=0, which is the first radar acquisition of the day (01:00). VWCt - VWCt0 simply describes the difference between VWC at time t and VWC at time t=0 (01:00). So the magnitude scaling of VWC does not affect the derivation of the b-parameter in eq. 6.

The reason it is valuable to use our destructive samples for scaling in Figures 8 and 9, is to be able to evaluate whether the diurnal cycles are more or less reasonable. Due to the destructive samples, we for example know that the reconstructions on June 9 and/or 10 would lack refilling in the evening if we wouldn't use the weighted average between forward and backward reconstruction, since significant gaps arise between reconstructions and samples (see Fig. 8). If a multi-day period is analyzed, such as in Figures 8 and 9, we advise to still use regular destructive samples for scaling, particularly for a fast-growing crop like corn.

(12) Line 235: I was worried about using P-M equation and CDF matching to sap flow to estimate transpiration because transpiration is very hard to estimate/measure. However, Fig. 5 shows this generally works well. It is stated somewhat indirectly, but I would emphasize clearly here or elsewhere that Fig. 5 shows that while modeling transpiration is a major drawback of the method, it works generally well in representing the VWC diurnal cycle.

Response: Addressed this in the Discussion, lines 374 - 375, which now read "While the indirect estimation of transpiration could be considered a drawback of the method, Fig. 6 has shown that the diurnal VWC cycle was represented generally well."

(13) Line 245-247: Does this mean there is evidence that full rehydration does not take place overnight every day and that capacitance is large enough to have some storage deficit carry-over from day to day? And that the assumption to use sum of sap flux over the day does not hold (lines 144-146)?

Response: Indeed, this shows that the assumption that all withdrawn water is exactly replaced within 24 hours does not hold for most days. There can be several explanations. From our predawn destructive samples (depicted in Fig. 2, 8 and 9), we for example know when there are increasing VWC trends as a result of growth, and decreasing VWC trends as a result of senescence. In these situations, the 24-hour assumption does not hold. Similarly, the effect of a dry period can result in decreasing VWC trends, related

to hydraulic capacitance of the plant. In this case, the 24-hour assumption does not hold either.

(14) Line 266: I think Fig. 8 is somewhat of a disservice to the authors and their nice results. The approach is well set up for sub-daily sampling, but ad-hoc modifications like CDF matching and VWC scaling are needed to represent day to day variations. I would say that the method is strong and well-developed for evaluating sub-daily VWC variations and a bit weaker for evaluating daily variations. In Fig. 8, my eyes are drawn more to the daily than diurnal changes which is not the focus of the paper and section heading. Consider showing diurnal variations individually for a few days (by segmenting individual days) to emphasize the results if possible.

Response: Thanks for the suggestion. We have added Figure 1 below to the manuscript, showing zoomed-in diurnal variations of June 7, 9 and 11.

[Figure]

Figure 1: Zoomed-in diurnal variations of backscatter and moisture for June 7, 9 and 11 (2018).

(15) Line 292: Is this unexpected that VV is more sensitive than cross-pol to vegetation?

Response: Note from Table 2 that the differences in sensitivity to VWC between VV and cross-pol are small. On June 9 for example, VWC changed with 0.5 kg m$^{-2}$, which would translate to a change of 1.5 dB (VV), and 1.2 dB (cross), if soil moisture and SCW would be stable. With the (inputs of the) regression not being perfect (see discussion), one should not put too much weight on this difference. The modelling study in Vermunt et al. (2020) showed that both VV and cross-pol backscatter were dominated by the vegetation/volume scattering contribution in this period. Given that most of the water is in the stems (Vermunt et al. (2020)), which are vertical structures, it is not surprising that vertically polarized backscatter is sensitive to dynamics in the vegetation.

(16) Line 300: Dew is receiving increased interested in its impact on diurnal observations of microwave emission and backscatter. Can the authors contextualize the dew results in the table a bit more? It is hard to tell if "c" is a large or small contribution to the signal compared to "b" without knowing typical dew variations in kg/m2. Maybe a variance-explained or normalized slope metric can help readers determine how much dew and internal water content relatively influence each backscatter signal. Only comparing the absolute slopes here does not fully show the relative contribution to the signals. It seems the authors are

exhibiting less confidence in the dew results (i.e., lines 371-377) and it is not clear why (while the result in lines 366-370 are very interesting!).

Response: Maximum diurnal dew estimates ranged from 0.04 to 0.46 kg m-2 in this period (Figure 9), with an average of 0.24 kg m-2 (Figure 11). Note that these are estimated quantities (Vermunt et al. (2020), and are similar to findings from Kabela et al. (2009). In lines 353-359, we describe how coefficients for soil moisture, VWC and dew translate to changes in backscatter for a typical dry day during the campaign of 2018 (June 9). It is true that we exhibit less confidence in the regression coefficients for dew, compared to those for VWC and soil moisture. First, this is because when we visually inspect Fig. 12, we see that nocturnal increase (as a result of dew) is barely visible, while variations due to VWC and soil moisture are represented quite well. This suggest that the 'c' coefficient underestimates the effect of dew on backscatter. Second, Table A4 shows that, for all polarizations, the P-values for SCW are higher than those for VWC and soil moisture. Nonetheless, with the exception of HH (P$>|t|$=0.286), all P-values for SCW are $< 0.05$, indicating statistical significance. So yes, we are confident that the regression is largely reliable, but we think that the effect of dew is underestimated. We think this could be improved if the estimates of SCW (and VWC) improve. Besides, SCW is not considered in backscatter models yet, so the relationship between dew and backscatter is not well known. We agree that this is a topic that requires urgent attention, and we are working towards including SCW in EM models. Addressed in lines 443-448, which now read: " However, it seems that the SCW coefficients (c) for VV and cross-pol in Table 2 underestimate the effect of dew on backscatter, as the nocturnal increases in calculated $\sigma_{VV}^0$ and $\sigma_{cross}^0$ in Fig. 12 are lower than observed. This could partly be addressed by improved SCW estimates, for example through inclusion of more leaf wetness sensors distributed in the canopy (Vermunt et al. (2020)). Moreover, additional research is needed to provide more insight into the scattering mechanisms under the presence of SCW, for example by considering SCW in physical backscattering models. "

(17) Line 308: Arguably, the destructive samples may be optional here, especially for sub-daily variations, which strengthens the results.

Response: Agreed. Changed sentence to "Our reconstruction results confirm that it is possible to estimate 15-min variations in corn VWC with only sap flow sensors and a weather station."

Line 400: I think it is worth mentioning the sapfluxnet project and how one can use those data (along with station or flux tower data) to validate time dynamics of VWC seen by passive and active satellites at large scales.

Response: Thanks for the suggestion. Addressed this, together with the suggestion in comment (20) in lines 470-476, which now read: "As radar observations are increasingly used to study plant water status, the presented sap flow method is a promising way to validate sub-daily satellite observations with just meteorological data and sap flow sensors, without laborious sub-daily destructive sampling. The method is expected to be most robust when the temporal resolution of the sap flow and ET observations are significantly smaller than the phase difference between the two, which depends on the species. The number of sensors required to capture VWC variations at footprint scale is expected to depend on the footprint size, and the spatial heterogeneity of vegetation type and factors influencing moisture supply and demand. Potentially, global database networks for sap flow measurements, i.e. *Sapfluxnet* [1], and flux tower measurements, e.g. *Fluxnet* [2] and *Ameriflux* [3] can play an important role here."
* * *
[1]http://sapfluxnet.creaf.cat
[2]https://fluxnet.org/
[3]https://ameriflux.lbl.gov/

(19) I recommend commenting on whether such an approach here can be used to evaluate day to day variations in VWC. Are there ways in which the transpiration = sap flow scaling in the early morning can be relaxed such that total storage over a day can be computed and evaluated day to day (line 147-149 start to get at this)? I know this becomes uncertain due to phase lags between transpiration and sap flux caused by the capacitance that the method is trying to measure. But I think this study is a nice step towards that and the authors recommendations for how that can be done or recommendations against it could be helpful for the active and passive microwave vegetation community. If the authors feel that is off topic, feel free to ignore.

Response: The use of multiple days for rescaling transpiration already allows for day-to-day variations. This means that total storage over a day can be estimated. However, Fig. 9(d) shows that the diurnal VWC reconstructions are discontinuous between consecutive days. The gaps indicate that the method is not perfect, and we recommend to test the suggested adjustments (lines 404-412) to improve the method before using it for the purpose of evaluating day-to-day variations in VWC.

(20) I also recommend commenting somewhere on how one could use such a method to validate satellite observations. What would be required in this case to estimate a diurnal VWC cycle at the footprint scale? It seems like a few meteorological measurements and sap flux sensors would suffice to at least understand the diurnal cycle.

See response on comment (18).

**References**

Guo, J.S., Hultine, K.R., Koch, G.W., Kropp, H., Ogle, K., 2019. Temporal shifts in iso/anisohydry revealed from daily observations of plant water potential in a dominant desert shrub. New Phytol. https://doi.org/10.1111/nph.16196

Holtzman, N., Anderegg, L., Kraatz, S., Mavrovic, A., Sonnentag, O., Pappas, C., Cosh, M., Langlois, A., Lakhankar, T., Tesser, D., Steiner, N., Colliander, A., Roy, A., Konings, A., 2021. L-band vegetation optical depth as an indicator of plant water potential in a temperate deciduous forest stand. Biogeosciences 18, 739–753. https://doi.org/10.5194/bg-2020-373

Momen, M., Wood, J.D., Novick, K.A., Pangle, R., Pockman, W.T., McDowell, N.G., Konings, A.G., 2017. Interacting Effects of Leaf Water Potential and Biomass on Vegetation Optical Depth. J. Geophys. Res. Biogeosciences 122, 3031–3046. https://doi.org/10.1002/2017JG004145

Konings, A. G., Saatchi, S. S., Frankenberg, C., Keller, M., Leshyk, V., Anderegg, W. R., ... and Zuidema, P. A. (2021). Detecting forest response to droughts with global observations of vegetation water content. Global change biology, 27(23), 6005-6024.

Vermunt, P. C., Khabbazan, S., Steele-Dunne, S. C., Judge, J., Monsivais-Huertero, A., Guerriero, L., and Liu, P.-W., 2020. Response of Subdaily L-Band Backscatter to Internal and Surface Canopy Water Dynamics, IEEE Transactions on Geoscience and Remote Sensing, 59, 7322–5907337, https://doi.org/10.1109/TGRS.2020.3035881.

Kabela, E. D., Hornbuckle, B. K., Cosh, M. H., Anderson, M. C., Gleason, M. L. (2009). Dew frequency, duration, amount, and distribution in corn and soybean during SMEX05. Agricultural and forest meteorology, 149(1), 11-24.

---

## Author Comment (AC2)

Reviewer comments in black
Reply to comments in blue

This research demonstrates the estimation of continuous vegetation water content (VWC) in corn crops at two research sites by adapting an existing method for measuring internal VWC in trees. Sub-daily VWC was succesfully calculated based on the difference between modelled transpiration and sap flow rates at the base of corn stems and constrained and validated with destructive sampling. Second, the research demonstrates the effect of diurnal variations of VWC and dew on radar backscatter. The study is innovative and is a valuable contribution to the field as it provides new methods and insight in current questions in microwave remote sensing, such as the effect of internal VWC and surface canopy water on the radar signal. I highly recommend to publish the paper, but I have some minor comments.

Response: Thanks for the careful consideration of the manuscript and the constructive comments. Below we have addressed the comments in blue. The line numbers in our replies refer to the revised manuscript.

(1) I believe the data and methods can be described a bit better. If I understood correctly, backscatter data is only available for the 2018 campaign in Florida, but sub-daily destructive samples are only available for the 2019 campaign in the Netherlands. So in short, the method to calculate sub-daily VWC is developed and validated on the 2019 data and then applied to the 2018 data to assess the effect of VWC on sub-daily backscatter variations. I think the paper will be easier to follow if this is stated clearly in section 3. I would even suggest to split data and methods in different sections for clarity.

Response: We agree that the use of data from the two campaigns was not described clearly enough early in the paper, and that this could have led to confusion. Therefore, we have now addressed this in the last paragraph of the introduction. The specific sentences in the introduction now read: "An extensive data set from a field campaign in the Netherlands in 2019 was used to evaluate the adapted method against diurnal cycles of VWC obtained by destructive sampling. Finally, the technique was applied to reconstruct sub-daily VWC variability of multiple consecutive days from another field campaign in Florida in 2018. In this campaign, high temporal resolution tower-based polarimetric L-band backscatter was collected. The reconstructed VWC was used, together with simultaneously collected soil moisture, surface canopy water (SCW), to gain better understanding of what controls sub-daily backscatter behaviour.". Moreover, we reorganized the methodology section in such a way that the two objectives of this study and corresponding data sets used are clearly separated now. We repeated which data sets were used to address which objectives in the first sentences of the new Data and Methods section, which now read "Section 3.1 relates to the adjustments and data required to make the sap flow technique (section 2) applicable to corn. Data from a field campaign in The Netherlands in 2019 were used to evaluate the adjusted method. Section 3.2 relates to the methodology and data used from our field campaign in Florida in 2018 for interpreting sub-daily backscatter behaviour. "

(2) Figure 2: make the colors more intuitive, either by making the colors an indicator for drought stress (e.g. brown/red), or otherwise a colormap according to date to make it easier to interpret.

Response: Done. We have changed the colors and used a colormap according to date.

(3) Section 3.2.1: It is unclear on which days the destructive samples were taken. It can be seen in figure 2 (but here there are only 7 days, whereas section 3.2.1 states 14 days?), but I think this information should be mentioned already here. To me it led to some confusion at line 230 in combination with section 3.2.2 where it states that because of power issues when measuring sapflow only three days have all data needed to estimate and validate VWC: July 25, Aug. 23 and 28. A table with an overview of days with destructive samples and sapflow data yes/no would be informative.

Response: We included a Figure in the revised version of the manuscript with an overview of the periods for which sap flow, ETo and sampled VWC are available: Fig. 1.

(4) Line 243: On July 25 all available data for the CDF-matching were used. Why? What is the difference with the other days?

Response: Each panel in Fig. 5 shows the best VWC estimate for the particular day, given the particular rescaling method (linear or CDF-matched). We considered the best VWC estimate the one which has the best fit, i.e. the smallest RMSE between observed and reconstructed VWC. On July 25, the lowest RMSE was found when the CDF-matching was performed with all available 15-minute observations for sap flow and ETo. For August 23 and 28, better fits were found when a smaller subset of data was used for CDF-matching.

(5) Line 277: "A sharp backscatter increase after rainfall was observed in all polarizations". Yes this seem true for those rainfall events where soil moisture is also increasing strongly. The event on June 12th seems different, where CW increases significantly, but soil moisture shows a very small response. Here VV, HH and crosspol backscatter drop strongly, and then go back to the level before the event, or get slightly higher. Can you explain what is happening here?

Response: Please see the zoomed-in situation of June 12 only in Figure 1 below. Actually, we do see that the backscatter increase in all polarizations perfectly corresponds to the increase in rainfall interception (SCW). The backscatter drop between 9:30 and 13:00 can be explained by a drop in VWC, which was clearly visible in all other days but June 12 (see Fig. 9 in the manuscript). It seems that the VWC reconstruction on June 12 was not that good. The fact that the backscatter returned more or less to the level before the event could be due to a combination of slight soil moisture increase, and refilling of the plant's internal water storage (VWC).

(6)Figure 8: maybe only show those days you actually used for the fit?

Response: We think it is valuable to show June 4, 5 and 6 as well in Figure 8 and 9. Despite that VWC reconstruction on June 5 and 6 was considered less reliable, the other data sets (soil moisture and SCW) give us valuable information about backscatter variations on these days. Besides, June 4 contains a dense, reliable data set, including two samples. Morover, showing June 5 and 6 provides insight in which situations VWC reconstruction does not work well, which could aid improvements in future work.

(7) Figure 9: for the fit you consider the VWC of June 5 and 6 not reliable enough. But for fig 9 a and d are aggregated over 9 days. This means that you did use june 5 and 6 for figure 9, is this correct?

Response: This is correct. The reason why June 5 and 6 are included in Figure 9 is because we explicitly look

at the period midnight to early morning here. In this period, VWC is generally stable. We particularly do not trust the VWC reconstructions in the afternoon-evening on these days, because we see some physically implausible features in the time series there (see lines 315-318). But there is not much reason to doubt the estimated variability of midnight-morning VWC. Excluding June 5 and 6 here because of VWC would also exclude the valuable SCW, backscatter and soil moisture data. That is why we chose to not remove June 5 and 6 from the aggregated data plotted in Fig. 9 (a) and (d). For the regression, we actually needed reliable full days of data. The VWC reconstructions on June 5 and 6 could not meet that condition.

(8) Line 285: delete "the"

Response: Done

(9) Line 286: During the last four aggregated acquisitions... which are these?

Response: The acquisitions at 19:00, 19:30, 21:00 and 21:30. Added ... 'between 19:00 and 21:30' ...

(10) Line 295 and onward: where do the values for changes in soil moisture, VWC and SCW come from for typical dry days? Also the multiple linear regression to assess the effect of moisture stores on backscatter is somewhat unclear. I think it might be more sophisticated to do this calculation with all units in mm? It needs an assumption on soil depth and penetration depth, but it should be possible. If not, I think the statements in line 295 and onward are confusing, since sensitivity and mentioning e.g. "three times more sensitive" is not really the right term here since they units are not the same. Maybe change to something like: Note that the coefficients from soil and vegetation water stores (Table 1) have non-homogeneous physical units. Nonetheless, it shows us that for a typical dry day during the campaign of 2018, e.g. such as June 9th, soil moisture reduced with 0.02 m3m-3 and that this translates to a -0.5, -0.8 and -0.8 dB change in VV, HH and cross-polarized backscatter. During the same day VWC changed with 0.5 kg m-2, which would translate to a change of 1.5, 1.1 and 1.2 dB. This shows us that typical diurnal variation in VWC leads to a three times higher change in VV-polarized backscatter than a typical diurnal change in soil moisture.

Response: The suggested text that the reviewer gives entails exactly what we tried to say. Therefore, we adjusted the text in the manuscript in lines 352 - 359, which now read: " Nonetheless, these coefficients indicate that for a typical dry day during the campaign of 2018, e.g. June $9^{th}$, the soil moisture reduction of 0.015 $m^3m^{-3}$ translates to a -0.4, -0.6 and -0.6 dB change in VV, HH and cross-polarized backscatter, respectively. During the same day, VWC changed with 0.5 kg $m^{-2}$, which would translate to a change of 1.5 dB (VV), 1.2 dB (HH) and 1.2 dB (cross). This indicates that on this typical dry day, a diurnal variation in VWC leads to an almost four times higher change in VV-polarized backscatter [dB] than a diurnal change in soil moisture does. On the same day, the changes in HH- and cross-polarized backscatter [dB] were two times higher for the diurnal VWC variations than for the soil moisture drydown.".

(11) Figure 10 is not discussed much in the text. It shortly states that the effect of SCW on backscatter is underestimated based on Fig. 10, but more explanation here would be good.

Response: Addressed in lines 360 - 366, which now read: "Fig. 12 presents the results of using the regression coefficients (Table 2), and the time series of VWC, SCW and soil moisture, to describe diurnal variations in backscatter. Each day is constrained by the first radar observation of the day, at 01:00. Note from the $R^2$ values in Table A4 that 68-71% of the variance in backscatter is explained by the three predictors. The P-values for SCW are always higher than those for VWC and soil moisture. Nonetheless, with the exception of the SCW coefficient in the case of HH-backscatter ($P > |t| = 0.286$), all P values are < 0.05, indicating statistical significance. However, note from Fig. 12(a) and (c) that the observed nocturnal backscatter increase as a result of dew is barely visible in the calculated backscatter. This suggests that the regression underestimates the effect of dew on backscatter."

(12) I suggest to split section 5.1 in two at line 330. One deals with the development and validation of the method with in situ data. The second part is applying the method to a longer period and a different region.

To make a distinction between the two parts, we added a blank line. Besides, we changed the first sentences of each parts, which now read: "We tested the potential of a non-destructive sap flow approach to estimate sub-daily VWC variations in corn with data from our 2019-campaign...." (line 372-373) and " When the methodology with CDF-matching was applied to the 10-day period from our 2018 campaign, ... " (line 399).

(13) Line 310: what about August 23rd?

Response: See the evaluation of August 23rd in lines 385-387 and 390-396 of the revised manuscript.

(14) Line 352: Also here, i think using "1.5 to 3 times more sensitive" is not the right wording.

Response: Addressed this in lines 419 - 421, which now read "Our regression analysis suggested that, on a typical dry day, the diurnal cycle of VWC led to a two (HH- and cross-pol) to almost four (VV-pol) times higher change in backscatter than the soil moisture drydown did."

(15) Line 366 and onward: make more clear in the text what the results are from your study. Now it is hard to discern if these results are from another study or yours.

Response: Added references to Table 2 and Figures 9, 10 and 11 to make this clearer.

[Figure]

Figure 1: Full polarimetric L-band backscatter and separated effects for a June 12 (2018), with (a) VV-polarized scattering coefficient, (b) HH-polarized scattering coefficient, and (c) averaged VH and HV-polarized scattering coefficients, (d) reconstructed VWC and total canopy water, which is the sum of reconstructed VWC and SCW, and (e) soil moisture at 5 cm depth.

---

## Author Comment (AC3)

Reviewer comments in black
Reply to comments in blue

   This paper provides an update on a previous analysis (Vermunt et al. 2020) of microwave radar data taken in Florida in 2018 over a corn field. In this previous paper, the authors have identified a diurnal cycle in backscatter which may be related to changes in vegetation water content (VWC). However, validating this hypothesis requires sub-daily measurements of VWC changes which are notoriously hard to obtain. The authors thus present a technique to reconstruct daily changes in VWC from a combination of sapflow measurements and weather-station based estimates of evapotranspiration. They evaluate this technique against a set of sub-daily destructive VWC samples taken in another location. The technique is then applied to the 2018 Florida data and used to demonstrate that sub-daily changes in backscatter are consistent with the reconstructed diurnal variability in VWC (in addition to surface canopy water and soil moisture).

Considering what the authors aim to achieve, the study set-up and the available measurements are not 100% ideal. The absence of more reliable ET data (i.e. from a flux tower) is a bit unfortunate, as is the fact that only few days have all types of measurements available. Contrary to what may be thought from the title, the proposed technique is not able to entirely reconstruct VWC variability, rather it can be used to extrapolate sub-daily VWC behavior from a single measurement (made daily, for example in the morning). Still I believe this to be a very useful attempt, especially if one focuses on sub-daily variability alone, and it may guide future similar research. There is certainly an interest in reconstructing sub-daily VWC from fewer of the time-consuming destructive samples.

Response: Thanks for the careful consideration of the manuscript and the constructive comments. Below we have addressed the comments in blue. The line numbers in our replies refer to the revised manuscript.

I have a few comments below which I think need to be considered, followed by some more minor comments and suggestions.

Major comments —

(1) Figure 10. The presentation of this figure is a bit misleading. If I understood correctly, the regression only attempts to predict intra-day variability in backscatter (Eq. 3). The initial backscatter value for each day is not reconstructed, but taken from the measurement directly. This is why there is a perfect match between 'observed' and 'calculated' at the start of each day. This should be made much clearer so as to not give the impression that the substantial inter-day variability in Fig. 10 can be explained from the regression. In fact, the quality of the regression for intra-day variability remains to be demonstrated as the authors do not report it (neither do they report if the coefficients of the regression are statistically significant).

Response: We agree that this was not clear enough in the manuscript. The observations used to constrain the predictions of sub-daily $\sigma^0$-variability , $\sigma^0_{t0}$, are now accentuated with open markers in Figure 12 (former Figure 10), and a description is added to the caption. Table A4 has been included to provide the reader with additional details on the regression and the statistical significance of the coefficients. The following text has been added regarding Figure 12, in lines 362 - 364: "The P-values for SCW are always higher than those for VWC and soil moisture. Nonetheless, with the exception of the SCW coefficient in the case of HH-backscatter (P> $|t|$=0.286), all P values are < 0.05, indicating statistical significance."

(2) In view of this, it's hard to tell if the regression is actually reliable, especially since much of the sub-daily variability in backscatter doesn't seem to be well predicted in Figure 10 (but it's hard to evaluate). Showing a scatter plot of the measured vs predicted sub-daily variations would be more informative in that respect.

Response: The $R^2$ values in Table A4 show that 68-71% of the variance in backscatter is explained by the three predictors. In addition, the P-values (with the exception of SCW and HH-pol) indicate that the regression is reliable. See previous comment.

(3) One could also make it clear which points are the ones that are used as the "anchor points" at $t_0$, for instance by giving them a different symbol color or shape.

Response: In response to your suggestion, we have changed the observations at $t_0$ into open markers, and added a description in the caption.

(4) Also the data in Figure 9 d-e-f provides the opportunity to better illustrate the modeled diurnal impact on backscatter (and compare it against the data in panels a-c). The contributions of all variables are mixed up in Figure 10, so it's difficult to learn much from that figure alone.

Response: We do not think that the data in Fig. 11 d-e-f (former Fig. 9) would better illustrate the diurnal impact from VWC, SCW and soil moisture on backscatter than the data used in Fig. 12 (former Fig. 10). We can discern the same periods from Fig. 11 in Fig. 12, but for all days individually. Fig 9 (d-e) can be used to analyse periods in detail. Using Fig. 12 in its current form will point a reader to some other interesting features too, such as the representation of backscatter increase after rainfall (June 8, 10 and 12) and the impact of a poor VWC reconstruction (e.g. June 12) on the backscatter simulations. Therefore, we chose to not replace Fig. 12.

(5) Section 3.2.3 is a bit difficult to read because the purpose or context of some new methods that are explained there only becomes apparent or fully understandable later in the paper. Maybe there is potential to reorganize this section a bit and potentially already illustrate the different approaches with a figure (Figure 4 provides some of that but too late for the reader). In general, the methods (when they document a new approach) seem a bit excised from the rest of the text. It wouldn't hurt to give a bit more meat to it, for instance by providing a figure to explain the reconstruction method in 3.3. as well (for instance, Figure 4 does that well for CDF-matching).

Response: In response to this comment, as well as comments from the other two reviewers, we reorganized the methodology section. In the new *Data and Methods* section, we have moved the text related to rescaling ETo from former 3.2.3 to new *3.1.1: Adjustments of the methodology.* In this new section, we merged all adjustments of the methodology presented in section 2 to make it applicable to corn. The different approaches to estimate transpiration are highlighted in a new table: Table 1. Moreover, we added an extra panel to Fig. 5, which shows the effect of the three approaches on the transpiration estimate. We chose not to add Figures containing our data at this stage of the paper, because the data collection is described in section 3.1.2. Instead, we added a high-level summary of the steps taken to reconstruct diurnal VWC cycles in the last paragraph of section 3.1.1, which now reads: "In summary, we adapted and evaluated the sap flow methodology to estimate diurnal cycles of corn VWC through the following three steps.
① The diurnal cycle of transpiration was estimated from ETo and sap flow data, using three different approaches (Table 1).
② Sub-daily variations in VWC were estimated by calculating the cumulative difference between 15-minute basal sap flow and transpiration estimates (eq. 1).
③ The resulting estimates of diurnal VWC variations were compared against destructive measurements of VWC." Please see also the reply on comment (10) for details about the different approaches to estimate transpiration.

(6) Are there any downsides to CDF-matching? You force the T rates to follow the same distribution as the sap-flow rates. Is there any evidence that this is may or may not be true in papers comparing transpiration and sap flow measurements? I think it's fine to test this method, but the implications and plausibility should be better discussed. For instance, there is a physical rationale for having a long-term balance between sap flow and T rates that justifies the 24-hour (or more) sum approach.

Response: The long-term balance between sap flow and T rates still holds for CDF-matching. That is not different from the linear approaches. See from the example in Fig. 1 below (in response to comment 10) that the distribution of ETlinear3d and ETcdf3d is quite different, but the 3-day sums are 17.04 mm and 17.07 mm, respectively. Based on the plant hydraulics theory described in section 2, lines 72 –75, it makes more sense that sap flow follows transpiration with some time lag, with similar peaks and during a similar period of time (e.g. Fig. 4(e)). From this point of view, there is no physical rationale for the distribution of the linear approach, with sap flow having much higher peaks than transpiration, during a shorter period of time (e.g. Fig. 5(b)). In fact, earlier experiments suggest that the diurnal distribution of sap flow and transpiration are actually quite similar (Miner et al., 2017). This is something we tested through CDF-matching, and it turned out that CDF-matching gave the best fit between sampled and modelled diurnal VWC cycles.

Minor comments —

(7) Title: because the proposed method still requires some daily VWC measurements as constraints. I wonder if "Extrapolating continuous vegetation water content . . ." would be more appropriate and a better description of the paper's contribution. Alternatively, you could put the emphasis on sub-daily ("Reconstructing diurnal vegetation water content...") , which does not need daily VWC measurement as constraint if one focuses on anomalies.

Response: Agreed. Changed title to 'Extrapolating Continuous Vegetation Water Content to Understand Sub-daily Backscatter Variations'

(8) L49: unavoidable suggest to replace with acceptable

Response: Agreed. Changed to 'acceptable'.

(9) L83: Was a bit hard to get on first read. Maybe modify the sentence into: "... lag between transpiration and upper sap flow, compared to the lag with basal sap flow, ...".

Response: Agreed and modified.

(10) L145-155: It may be useful to provide an illustration of the time series (before and after correcting ET with these different processing options) as a supplementary figure. Right now, it is a bit difficult to visualize what is happening to the ET time series. By the way, even if P-M ET was a perfect method and produced close to truth ET time series, you'd still need to separate the plant transpiration part from the soil evaporation part. My point is that the "correction" actually also serves to do that operation.

Response: Figure 1 below (also added to supplementary materials) gives an example of the effect of the three rescaling methods. What stands out is that the cdf-matched rescaling provides significantly higher peaks, compared to the linear rescaling. But in this case, the 3-day sum of $ET_{cdf-3d}$ is not that different from the 3-day sums of $ET_{linear-24h}$ and $ET_{linear-3d}$: 17.07 mm, 17.35 mm, and 17.04 mm, respectively. This is because below 0.12 mm/15min, cdf-matched ET is lower than linear rescaled ET. An extra panel (a) was added to Fig. 5 in the manuscript, which illustrates the effects of the three approaches to estimate transpiration from ETo and sap flow. Explanatory text was added to section 4.2, which reads: "Fig. 5(a) illustrates the effects of the three approaches to estimate transpiration from ETo and sap flow (Table 1). T-cdf and T-3d represent the CDF-matched and linear estimates of transpiration, for which 3 days of data were used: July 24-26. What stands out is that the CDF-matched rescaling (T-cdf) provides a significantly higher peak, compared to the linear rescaling (T-24h and T-3d). On the other hand, when ETo rates are 0.09 mm 15min$^{-1}$ or lower, T-cdf was lower than the linear estimates. Both linear transpiration estimates were close in this particular case, which means that the ratio of the 24h sum of sap flow over ETo was close to the ratio of the 3-day sum of sap flow over ETo."

[Figure]

Figure 1: ETo and the three rescaling approaches for July 24-26.

(11) L153: I thought on first reading that CDF matching was done with the daily totals (not the sub-daily time steps). This may need to be mentioned here.

Response: Addressed this by changing the sentence to 'This matching was achieved by first ranking all 15-minute data from both data sets from low to high values, ...'

(12) L155: It could be useful to give a final high level summary of what happens here. For instance: "information on the diurnal shape of ET is entirely derived from Penman-Monteith, but the ET daily totals

are scaled so that T estimates that are consistent with sap flow over long periods of time".

Response: Added "This means that information on the diurnal shape of ETo is derived from the Penman-Monteith equation, and that these ETo estimates are then scaled so that the resulting transpiration estimates are consistent with sap flow over a given period of time." to updated lines 113-114.

(13) Equation 2: I think the notation is not appropriate (or at least it is very unclear to me). I think I understand what you did in the end, but the equation does not reflect it: Is "k=15 minutes" meaningful here? The lower position should indicate the starting point (i.e. $k = t_0$, or $k = t_0+15$ minutes), check for instance: http://www.columbia.edu/itc/sipa/math/summation.html In Fk and Tk, does k denote the start or the end of the 15 minute time period? Why multiply (Fk – Tk) by $\Delta$t, if Fk and Tk are already expressed in per 15 minute rates? (I assume $\Delta$t would equal 15 minutes, since t and $t_0$ are indicated to be expressed in minutes).

Response: Thanks you for pointing out this mistake. This is addressed in lines 80-82, which now read: "

$$VWC(t) = VWC(t_0) + \sum_{i=t_0}^{t}(F_i - T_i)\Delta t \tag{1}$$

, where $VWC(t)$ is the estimated VWC at time t, $VWC(t_0)$ is a reference VWC at t=0, $F$ is basal sap flow, $T$ is whole-crown transpiration, both in mass per unit of time, and $\Delta$t is the duration of a time step."

(14) L188: Why these 10 days in particular?

Response: In the reorganized methodology section, this is explained in lines 199-200, which read "The longest period for which we had all data available was from June 4 00:00 to June 13 10:15.", and in lines 241-242, which read "The period of consecutive days for the analysis was limited by the availability of sap flow data. A 10-day time series was found in mid-to-late season which contained continuous sap flow and weather data, L-band backscatter, and five sampling days."

(15) 192: "did not overlap". I don't understand what this means. Do you simply mean, if they are not equal to each other?

Response: Indeed. Re-phrased the sentence, which now reads: "In case there was a gap between forward and backward reconstructions,...", see line 245-246 of the revised manuscript.

(16) L200: So this expression allows for an investigation of the sub-daily dynamics and basically removes the potential inter-day differences (since all data is relative to $t_0$). Maybe this should be stated more explicitly

Response: This is clarified by adjusting the text around the expression, which now reads: "The separate effects of the three different moisture stores on sub-daily backscatter ($\sigma^0$) variations were quantified through multiple linear regression. The relation between sub-daily backscatter variations and changes in these dynamic moisture stores was described by:

$$\sigma^0(t) = \sigma_{t0}^0 + a(\theta_t - \theta_{t0}) + b(VWC_t - VWC_{t0}) + c(SCW_t - SCW_{t0}) \tag{2}$$

,where $t_0$ is the first radar acquisition time of the day (01:00), and assuming linear relations between $\sigma^0$ and the individual moisture stores. The regression coefficients $a$ $[dB/m^3m^{-3}]$, $b$ $[dB/kgm^{-2}]$, and $c$ $[dB/kgm^{-2}]$ were used to quantify the change in backscatter within a day as a result of change in moisture, and were derived for each polarization separately. ".

(17) L219: It is unclear what is meant by "the linear estimate". I guess this means the scaling to match the 24-hr totals. Maybe section 3.2.3 needs to be better structured. You could potentially make a quick list of the different methods which you are testing and comparing.

Response: We indeed referred to the scaling to match the 24-hr totals as 'the linear estimate'. In the reorganized methodology section, we included a table (Table 1), which gives a clear overview of the three methods we compared and tested, including their assumptions and equations.

(18) L227: "observed [on that day] from"

Response: Added 'on that day' for clarification

(19) Figure 4. It is assumed that ET estimates need correction to maintain some balance between transpiration and sap flow, but what about biases in sap flow measurements for high rates of flow? Are they possible and how big could they be?

Response: Biases in the presented sap flow measurements for high rates of flow are unlikely, because first of all, the sensor installation with shield and proper insulation limits thermal noise from radiation or other effects. Moreover, the Dynamax programme uses a built-in high flow-rate filter to prevent a distortion of the accumulated flow over those rates that are reasonable (Dynamax, 2007). Possible extraneous observations from a single sensor in 2018 are levelled out by averaging four sensors.

(20) L242: "An exception to this rule was July 25, when all available data for the CDF-matching were used." I don't understand why this is an exception, which sample was used as a constrain there then?

Response: This sentence was omitted in the revised manuscript.

(21) Figure 5. In each time series, it would be useful to show with a different symbol the one sample VWC that was used as constrain.

Response: Agreed. We changed the symbol for the measurements which were used to constrain the reconstructed lines in Figures 5, 7 and 8, and included explanations in the captions.

(22) Figure 5. This Figure shows well how the 24-hour method does not allow for a difference between the start and end-of-day VWC. Could be mentioned.

Response: This is addressed in lines 299-301, which read "The upper row clearly shows that the linear-

24h approach does not allow for a difference between the start and end-of-day VWC, while the inclusion of multiple days does."

(23) Figure 5. Unlike the other days, Aug 23 had a lot of dew, so it could be that the VWC measurements were biased up because of that (one can remove dew with paper towels only on the accessible parts of the plant). This would explain why the reconstruction has a hard time for that day.

Response: Thanks for the suggestion. However, we do not think this can explain why the reconstruction is poor on Aug 23. Our sampling protocol involves removing the whole plant from the field, separating the leaves and stems and removing all dew with paper towels. So, we are confident that there is no bias due to residual dew due to inaccessibility. Furthermore, residual dew would lead to an overestimation of VWC. This would then particularly hold for the measurement at 6:30, because at 10:00 all dew was dissipated. A lower VWC at 6:30 (or even 10:00), would still result in a different shape of the sampled diurnal VWC cycle compared to the estimated diurnal cycle.

(24) L264: "see fig 4d". It's hard to understand how this relates to what is being said. This could be better explained.

Response: This was an error. It should have been a reference to Figure 8(a) (former Fig 7(a)), which shows that ET on June 6 is markedly different to that on the other days. For clarity, the sentence is reformulated in lines 316 - 318 , and now reads: "Despite the advantage of CDF-matching, opposed to linear conversion, to better reflect diurnal extremes, the anomalous dynamics of June 5 and 6 are not captured sufficiently.".

(25) Figure 9. It would be useful to show some +/- 1 std deviation error bars (or envelopes) around the averaged data.

Response: Please see Figure 2 below for the mean +/- 1 standard deviation. We think the added value of the standard deviation is low, and including them in the Figure will negatively affect the readability of the Figure. Therefore, we decided against adding them to the Figure.

(26) L285 typo

Response: Done

(27) L298: Is it 3 times more if the units are dB ? (and same later)

Response: Rephrased sentences with this statement here, and later, based on a comment from another reviewer. This specific sentence (lines 356-359) now read: " This indicates that on this typical dry day, a diurnal variation in VWC leads to an almost four times higher change in VV-polarized backscatter [dB] than a diurnal change in soil moisture does. On the same day, the changes in HH- and cross-polarized backscatter [dB] were two times higher for the diurnal VWC variations than for the soil moisture drydown."

(28) L301: I don't understand why (where?) Fig. 10 would show that. Please indicate what you mean

[Figure]

Figure 2: Mean and standard deviations of backscatter (VV, HH, cross-pol), VWC, SCW, and soil moisture for the periods described in Figure 11.

about Fig. 10 more clearly.

Response: When we look at VV-pol backscatter in Fig. 12 (former Fig. 10), we see that each night observed backscatter increases until the 7:30 acquisition (except from June 12). We have seen before that this increase can be attributed to dew formation, because VWC and soil moisture do not increase in these periods. Meanwhile, the calculated backscatter stays stable or only increases slightly. Similar patterns can

be observed for cross-pol. This suggests that the regression results underestimate the effect of SCW on backscatter. Rephrased this sentence in lines 364 - 366, which now read " However, note from Fig. 12(a) and (c) that the observed nocturnal backscatter increase as a result of dew formation is barely visible in the calculated backscatter. This suggests that the regression underestimates the effect of dew on backscatter. ".

(29) Table1: Was the significance of the coefficients tested? Please report if they are statistically significant, their confidence interval, and what is the overall performance of the regression.

Response: In the revised version of the manuscript, we included Table A4, which shows the results of the multiple regression analysis (see also responses on comments (1) and (2)).

(30) L310: "of dew" => "that dew"

Response: Changed 'of' to 'that'

(31) L317: "This is comparable to estimated dew evaporation in this period, which was 0.09 kg m2". Can you explain where this estimate comes from?

Response: See Fig. 7. The black line in the bottom left figure represents dew on July 25. The peak at 6:00 is 0.09 kg m-2 (see y-axis on the right). At 8:15, all dew was evaporated. Added "(Fig. 7)" to line 385.

(32) Does the temperature of the canopy water or of the soil water have any possible impact on backscatter and if yes, could it explain some of the diurnal variability?

Response: For the range of temperatures observed, the primary driver of variations in dielectric constant (of soil or vegetation) is water content, with water chemistry and temperature being of secondary importance. Temperature becomes highly significant when the water in the plant or soil freezes as the water is bound rather than free, resulting in a sharp decrease in dielectric constant as the temperature goes below freezing (Schwank et al. (2021); El-Rayes and Ulaby (1987)). A preliminary analysis of the data provided by El-Rayes and Ulaby (1987) (Figure 11) suggests that VWC effects dominate. However, the figure assumes that the sample did not dry out during the period in which the temperature was changed, and has few data in the temperature range we observed. We are not aware of experimental datasets that consider both temperature and moisture variations in vegetation, so this is something that warrants attention as sub-daily microwave data become available. Based on the results of Schwank et al. (2021), temperature seems to primarily become significant when freezing occurs.

(33) L340-345: Yes I think most of the re-scaling approaches you presented here would still be potentially needed to get from measured ET to T.

Response: Agreed. However, we expect that when diurnal E variations could be excluded from the ET measurement as much as possible, one can get better estimates of diurnal variations in T, and potential errors after re-scaling with sap flow may be smaller.

(34) L350-355: This is based on the fitted coefficients but it's not clear if these are actually significant.

Response: See response on comment (1).

(35) L357: I agree that it is a credible interpretation of Figure 9, however, I think it would be more convincing if a physical model of backscatter was there to demonstrate that both effects are indeed of similar magnitude and can cancel each other. But I guess this would also mean adding a whole new section to the paper..

Response: We agree that the use of a physical model is potentially a convincing tool to demonstrate these opposite effects on backscatter. However, widely used physical models are generally developed and calibrated based on seasonally variant VWC only. As a consequence, the effect of sub-daily VWC variations on backscatter are not captured very well (and SCW is not included at all). We are certainly keen to adjust physical models in such a way that they can handle both internal VWC and surface canopy water on sub-daily timescales. However, this is sufficiently complex to warrant a separate manuscript by itself.

(36) The conclusion makes a good summary and some good points on why the research is relevant, good job! It would also be interesting to read the authors' perspective on what type of future work would be needed to achieve better comparability between in-situ microwave data and eco-hydrological observations. In particular, it seems that when it boils down to sub-daily variability only, the time lag between sap flow and the transpiration estimate will control most of the VWC cycle. If it's really the case, the authors may provide some recommendations on the needed temporal resolution (already touched on L335, but could deserve more space).

Response: Thanks. We addressed this in lines 470 - 476, which now read: "As radar observations are increasingly used to study plant water status, the presented sap flow method is a promising way to validate sub-daily satellite observations with just meteorological data and sap flow sensors, without laborious sub-daily destructive sampling. The method is expected to be most robust when the temporal resolution of the sap flow and ET observations are significantly smaller than the phase difference between the two, which depends on the species. The number of sensors required to capture VWC variations at footprint scale is expected to depend on the footprint size, and the spatial heterogeneity of vegetation type and factors influencing moisture supply and demand. Potentially, global database networks for sap flow measurements, i.e. *Sapfluxnet* [1], and flux tower measurements, e.g. *Fluxnet* [2] and *Ameriflux* [3] can play an important role here."

**References**

Schwank, M., Kontu, A., Mialon, A., Naderpour, R., Houtz, D., Lemmetyinen, J., Rautiainen, K., Li, Q., Richaume, P., Kerr, Y., and Mätzler, C. (2021). Temperature effects on L-band vegetation optical depth of a boreal forest. Remote Sensing of Environment, 263, 112542.
* * *
[1] http://sapfluxnet.creaf.cat
[2] https://fluxnet.org/
[3] https://ameriflux.lbl.gov/

El-Rayes, M. A., and Ulaby, F. T. (1987). Microwave dielectric spectrum of vegetation-Part I: Experimental observations. IEEE Transactions on Geoscience and Remote Sensing, (5), 541-549.

Dynagage Sap Flow Sensor User Manual (2007). Dynamax Inc., Houston, TX, USA.

Miner, G. L., Ham, J. M., Kluitenberg, G. J. (2017). A heat-pulse method for measuring sap flow in corn and sunflower using 3D-printed sensor bodies and low-cost electronics. Agricultural and Forest Meteorology, 246, 86-97.